# How Long Do Model Patches Last? A Temporal Perspective on PortLLM

## Abstract

As large language models (LLMs) undergo regular updates through continual pretraining, the temporal reliability of downstream fine-tuning methods becomes increasingly important. Parameter-efficient methods, such as low-rank adaptation (LoRA), offer scalable solutions for task adaptations without requiring full LLM retraining. More recently, PortLLM has been proposed as a training-free patching mechanism that permits patch reuse over consecutive LLM releases. Although these training-free methods are appealing when full fine-tuning is impractical, their temporal reliability remains underexplored. Using PortLLM-style patches as a baseline approach, we conduct large-scale experiments and found that PortLLM patching exhibits a statistically significant performance decline over time, even when the task and neural architecture remain unchanged. Our findings reveal that patch performance degradation is a general and measurable risk when PortLLM is applied over an extended period. The statistical observation of the declining performance trends forms the foundation for our proposed forecasting algorithms, which estimate failure dates and test hypotheses about target-date performance failures. These forecasting algorithms rely on historical performance indicators without requiring downstream fine-tuning or access to original training data. Our framework enables downstream developers to anticipate failure and make informed decisions about when retraining is necessary, thereby supporting reliable and cost-effective LLM maintenance.

## 1 Introduction

Large language models (LLMs) have achieved strong performance across many tasks, including language (Brown et al., 2020; Liang et al., 2023), math (Shao et al., 2024; Wang et al., 2024a), and reasoning (DeepSeek-AI, 2025; Wei et al., 2022; Team et al., 2025). Frequently updated LLMs incur significant retraining costs, posing challenges for downstream developers seeking to adapt LLMs to specific tasks. To reduce the retraining costs, a growing body of work has explored lightweight fine-tuning/personalization methods (Hu et al., 2022; Khan et al., 2025; Houlsby et al., 2019; Zaken et al., 2022) that inject task-specific knowledge into LLMs without full retraining of base models. Among them, PortLLM (Khan et al., 2025) proposes a data- and training-free patching method for portability of patches across temporally evolved LLMs. PortLLM's patching achieves strong task performance in short-term transfer to successive LLM base-model releases without retraining. However, the long-term reliability of patching has not been systematically studied, potentially leaving developers reliant on downstream patched models that silently degrade. In this work, we use PortLLM as a baseline to explore the temporal reliability of patched LLMs by posing an open question: How long can PortLLM patching remain effective as the base model evolves?

Figure 1 illustrates a PortLLM deployment scenario over an extended period of time, and typical questions being asked concerning patching performance degradation. An upstream LLM vendor periodically (e.g., quarterly) releases an updated base model through continual pretraining. A downstream developer fine-tunes the initial base model $\theta_0$ and generates a PortLLM-style patch $\Delta\theta_0$. Due to cost or data limitations, the same patch is applied to subsequent base model releases $\theta_1$, $\theta_2$, and $\theta_3$ without retraining. Over time, task performance mildly degrades. This raises two research questions (RQs) motivated by practical deployment: (RQ1) From the developer's perspective, when will patching performance fall below a threshold (red dashed line)? (RQ2) From the business planning standpoint, will computing resources be required or not by a future date (blue dashed line)

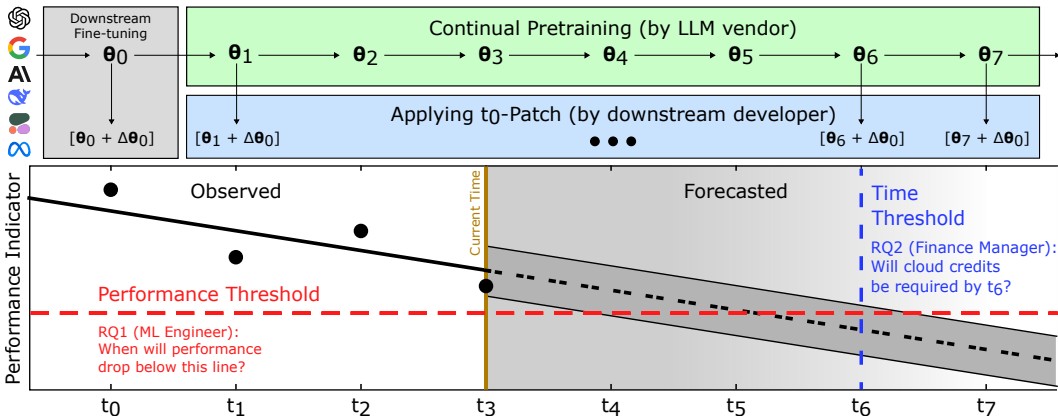

Figure 1: Overview of PortLLM deployment over an extended period of time and patch degradation forecasting. A downstream developer applies a patch $\Delta\theta_0$ trained at $t_0$ to a continually evolved base model $\Delta\theta_i$, $i \geq 1$, provided by an LLM vendor. As updates progress, the task performance of models $\Delta\theta_i + \Delta\theta_0$ degrades. Our forecasting algorithms use observed degradation trends to estimate future performance (shaded region) and answer two critical deployment questions of when and whether PortLLM patching will fail. Our framework enables proactive, cost-aware adaptation decisions without requiring retraining at every base model release.

for retraining?[1] Answering these questions requires modeling how patching performance changes over time and developing algorithms to forecast future performance based on observed trends.

Our work presents the first systematic study of temporal degradation in patched LLMs. We conduct extensive experiments on the Math Genie (Lu et al., 2024), BoolQ (Clark et al., 2019), ARC-Easy (Clark et al., 2018), and WinoGrande (Sakaguchi et al., 2021) datasets using LLMs such as Mistral-7B (Jiang et al., 2023). We track PortLLM-style patched model performance across successive base model updates and statistically quantify the rate and structure of degradation driven by patch misalignment resulting from continual pretraining. We model these patterns as a time series and propose statistical estimation and hypothesis testing algorithms to predict when and whether a patch will fail in deployment settings, respectively. In Appendix J, we offer theoretical analysis to explain the cause of patch performance degradation. Our contributions are threefold:

1. We conduct the first large-scale experimental evaluations on the temporal reliability of patched LLMs evolved via continual pretraining. While PortLLM (Khan et al., 2025) conducted a feasibility study of temporal patch portability, we provide statistically significant evidence that gradual performance decline occurs over time, even when tasks and architectures remain unchanged.

2. We perform statistical analysis of patching performance trends across model updates, quantifying the gradual degradation process. Building on these empirical observations, we develop a time series modeling framework that characterizes patching performance as a structured temporal process. This formalizes degradation dynamics as a measurable property and provides the basis for forecasting when adaptations are likely to fail.

3. Built on the modeling of degraded trends, we introduce lightweight, mathematically grounded algorithms for (i) failure date estimation and (ii) target date hypothesis testing to detect patch failure. Their principled monitoring of the temporal reliability of patched LLMs advises when or whether to retrain, without requiring retraining at every base model release.

4. Our theoretical analysis of patching performance degradation reveals it results from the divergence between the PortLLM patch and the "true" patch trained on the updated base model.

## 2 RELATED WORK

**Parameter-Efficient Personalization.** LLMs have achieved strong generalizability through scaling (Kaplan et al., 2020), instruction tuning (Ouyang et al., 2022), and reinforcement learning

---

[1]Retraining consumes computational resources, diverts engineers from other tasks, and may require domain-specific data that may be no longer available after a period of time. Forecasting the need for retraining enables teams to plan data access and allocate developer resources effectively, even if compute is not the primary concern.

from human feedback (Christiano et al., 2017), enabling extensive applicability across diverse tasks. Continual learning research (de Masson D'Autume et al., 2019; Sun et al., 2020) and continual pretraining studies (Jin et al., 2022) have further highlighted challenges arising from evolving corpora and distributional drift, primarily focusing on full-model adaptation across sequential tasks or datasets. While these advances enable broad task applicability, efficiently adapting models to specific domains or applications remains an open challenge. Parameter-efficient fine-tuning (PEFT) methods provide scalable alternatives to full-model adaptation by updating only a small subset of parameters. Low-rank adaptation (LoRA) (Hu et al., 2022), QLoRA (Dettmers et al., 2023), and related techniques use low-rank updating matrices to enable efficient adaptation. Adapter-based methods (Houlsby et al., 2019; Pfeiffer et al., 2021) insert task-specific bottleneck layers that allow for modularity and reuse. PortLLM (Khan et al., 2025) extends this line of research by injecting task-related patches into evolved base models without additional training. While these methods have demonstrated strong performance when evaluated on patched base models that evolved for a short duration, their effectiveness under extended, continually pretrained base models remains largely unevaluated. Our work studies the behavior of task-specific adaptations across many sequential LLM updates, focusing on whether PortLLM patches remain effective as the base model evolves.

**Drift Modeling and Temporal Analysis.** Performance degradation over time has been studied in several domains through the lens of drift detection and anomaly analysis. Classical work on concept drift (Gama et al., 2014) and unsupervised anomaly detection (Chandola et al., 2009) provides statistical tools for identifying failure trends in evolving systems. More recent efforts in dataset shift detection (Rabanser et al., 2019) and concept drift characterization (Webb et al., 2016) extend these ideas to machine learning. These methods focus on evolving input distributions, whereas our setting concerns parameter drift—continual pretraining shifts the base model so that a static PortLLM patch gradually misaligns, even under fixed tasks and inputs. This patch-based misalignment creates a qualitatively distinct failure mode as developers may unknowingly rely on patches that silently degrade. We offer more thorough analysis for how this misalignment affects performance in Appendix J. Our work innovatively applies temporal techniques to address the performance degradation observed in our large-scale experimental evaluations.

## 3 LARGE-SCALE EXPERIMENTAL STUDY AND EVIDENCE OF DEGRADATION

We aim to provide empirical grounding for whether the PortLLM patching method can maintain long-term utility. We specifically investigate the performance of applying the same PortLLM patch over time as the base model evolves through continual pretraining. We conduct multiple independent repetitions of continual pretraining to ensure that the results are statistically significant.

### 3.1 DATASETS AND EXPERIMENTAL CONDITIONS

**Datasets.** We evaluate patching performance on four benchmarks spanning mathematical reasoning, reading comprehension, and commonsense inference, namely, MathGenie (Lu et al., 2024), BoolQ (Clark et al., 2019), WinoGrande (Sakaguchi et al., 2021), and ARC-Easy (Clark et al., 2018) (see Appendix A for more details). Continual pretraining is performed on UpVoteWeb (UpV, 2024), a filtered Reddit corpus that reflects evolving internet discourse and is a minutiae proxy for web-scale mega-datasets curated for continual pretraining of base LLMs in production settings. Additional experiments were performed on a broader collection of datasets structured into a time series (see Appendix I) to ensure generality across pretraining sources. Pretraining on this broader collection introduces substantial distributional shifts across time steps, allowing us to model the temporal evolution of data even more radically than typically observed in real-world scenarios.

**Evolution of LLM's Model Parameters.** We simulate real-world evolution of LLMs by continually pretraining a base model in successive stages and tracking performance of LLM-patches over time. Specifically, we pretrain Mistral-7B (Jiang et al., 2023) on UpVoteWeb in 12 chronological two-week-long segments[2], yielding 13 model versions spanning five months. A single PortLLM patch $\Delta\theta_0$ is trained at timestep $t = 0$ and applied via addition in parameter space to all evolved base models $\theta_t$ at $t \in \{1, \ldots, 12\}$. This will isolate the effect of misalignment between a PortLLM-style patch and the base model, while keeping task, architecture, and patching procedure unchanged.

---

[2]We divided UpVoteWeb into equal token count segments. Appendix G shows that varying segment density yields similar degradation trends, indicating that our results are not sensitive to this choice.

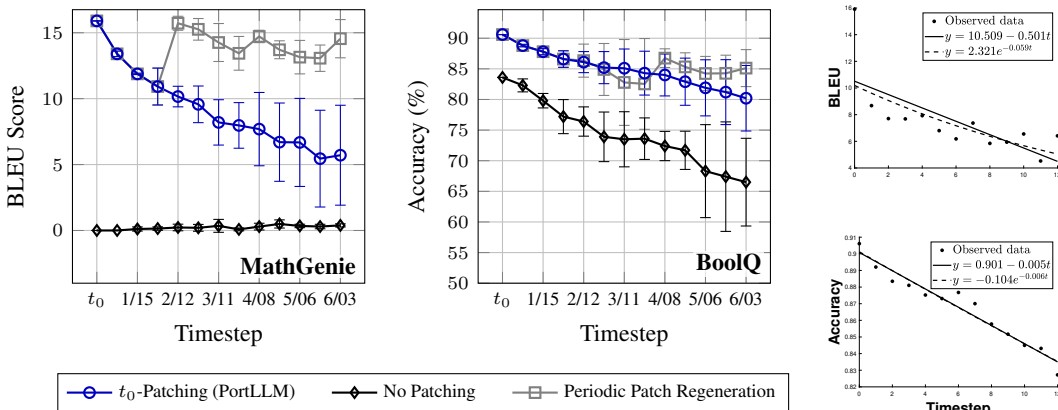

Figure 2: PortLLM patching performance (blue "○") for MathGenie and BoolQ over 13 temporal checkpoints continually pretrained on the Reddit dataset. PortLLM patching outperforms the no-patching baseline (black "◇") by a significant margin (as in Khan et al. (2025)), while our experiments reveal that its performance degrades steadily over time (Section 3). Error bars denote the standard deviation across 5 independent repetitions of continually pretrained base models. The two example plots on the side for MathGenie and BoolQ show linear and exponential curve fitting results for the PortLLM patching performance of a single repetition. Finally, periodic patching (gray "□") at every four time steps can boost performance at each refreshing step, but it may not be cost-effective if the downstream developer can predict when PortLLM patching fails (Section 4).

**Reproducibility.** Repetitions. All evaluations of PortLLM's patching degradation are repeated over 5 independent continual pretraining repetitions to ensure statistical confidence. Experiments were repeated with unique random seeds. Exact seeds and configuration files will be released with the codebase. Compute. All experiments were run on Nvidia A100 GPUs (40GB). Each continual pretraining timestep was distributed across 8 GPUs and took approximately 2 wall-clock hours. LoRA-based patch fine-tuning at each checkpoint required 15–45 minutes, depending on the task. LoRA Configuration. For continual pretraining of base models, we applied LoRA with $r = 64$ and $\alpha = 128$ to all attention submodules (i.e., query, key, value, output projection) and feed-forward layers (i.e., up, down, and gate projections, where applicable). For downstream fine-tuning, we used a LoRA with $r = 8$ and $\alpha = 16$ targeting the same set of LLM modules.

**Metrics.** We adopt the same evaluation metrics across all temporal checkpoints to ensure fair comparisons. All tasks are evaluated using the LM-Eval-Harness framework (Gao et al., 2024) with standardized decoding and scoring procedures. For MathGenie, we report the BLEU score (Papineni et al., 2002) to measure the structural fidelity of generated math explanations. While absolute correctness is difficult to verify at scale, BLEU provides a stable approximation of answer coherence. Following established evaluation conventions, we report test accuracy as the primary metric for BoolQ, ARC-Easy, and WinoGrande. These metrics enable us to track PortLLM's patching performance as the base model evolves.

## 3.2 EVIDENCE OF PATCHING DEGRADATION OVER TIME IN LARGE-SCALE EXPERIMENTS

We experimentally evaluate how PortLLM's patching strategy performs as the base model evolves through continual pretraining. The two larger plots in Figure 2 present results on MathGenie and BoolQ across 13 successive model checkpoints. A patch trained at timestep $t = 0$ (blue line with "○") initially provides substantial gains over the no-patching baseline (black line with "◇"). However, its effectiveness declines steadily with additional base model updates. Statistical analysis in Section 3.3 further supports this observation: Linear and exponential decaying trends significantly outperform constant/no trend across all benchmarks. While Figure 2 presents results on MathGenie and BoolQ, we observe similar patterns on WinoGrande and ARC-Easy in Appendix H. Despite differences in domain and problem characteristics, all benchmarks reveal that the benefit of $t_0$-patching erodes predictably as the base model evolves. This evidence across diverse datasets augments the preliminary finding from PortLLM (Khan et al., 2025), providing the first systematic experimental evidence that temporal degradation is not task-specific, but a broader limitation of the $t_0$-patching strategy applied to the continually evolving base LLMs.

We attribute the observed performance decline to the misalignment of the evolved base LLMs with the $t_0$-patch. From an optimization perspective, continual pretraining modifies the optimization landscape such that the descent direction guided by the $t_0$-patch may no longer align with the new optimal descent direction in the optimization landscape of the evolved base LLM (see Figure 29 for an intuitive, geometric illustration). Long-term evolutions of the base LLM can alter the optimization trajectory even more, resulting in much weaker adaptations. We provide a complete theoretical justification in Appendix J.

Our experimental observation motivates the development of adaptation strategies that explicitly account for temporal degradation. The gray curves with "□" in Figure 2 show the PortLLM performance when patches are retained every 4 time steps for restoring alignment. While this tactic of periodic patch regeneration can mitigate temporal degradation, it requires significant time and monetary overhead for medium-term deployment if regeneration is performed more frequently than necessary. Furthermore, it requires continual access to domain-specific data for patch retraining. Section 4 will present forecasting algorithms that indicate when intervention is necessary.

### 3.3 TIME SERIES ANALYSIS AND DETECTION OF PATCH PERFORMANCE DEGRADATION

The two example scatter plots in the right panel of Figure 2 confirm that the performance of PortLLM patching slowly degrades temporally. Both plots show observed data points (scatter points), a linear curve fit (solid line), and an exponential curve fit (dashed line). We selected linear and exponential parametric models based on our observation of the data that performance degradation is generally consistent without abrupt jumps. The exponential curves are obtained by performing a linear fit on log-transformed response variables (Faraday, 2005). The BoolQ data points shown in Figure 2 aligns closely with a linear model, whereas MathGenie shows greater variation but still follows the same slow decline. This supports modeling degradation with linear/exponential parametric trends. The use of a parametric model is further supported by the remaining curve fitting plots shown in Figures 10 to 15 of Appendix B. Tables 3 and 4 of Appendix B further provide detailed statistical results for testing a linear/exponential trend of patching performance against an intercept/constant performance. The $p$-values in Table 3, which shows results for pretraining on the Reddit dataset, provide statistically significant evidence with $p$-value $< 0.05$ against an intercept model in $49$ out of $50$ tests. The evidence in Table 4, which shows results for pretraining on the combined time-series dataset, is statistically significant in $16$ out of $22$ tests.

The combined time-series dataset introduces stronger temporal distributional shifts than typical in practice, yet we still see statistical significance, indicating our models may remain robust under realistic temporal drift. For the time ranges where the model performance should be high enough to be practically useful, linear trends sufficiently model the data for the algorithms in Section 4.

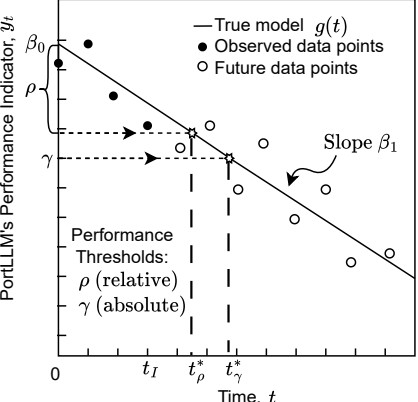

## 4 PROPOSED DEGRADATION FORECASTING ALGORITHMS

To identify an LLM's downstream retraining needs, we propose two forecasting algorithms to facilitate technical and business planning, respectively. First, we use a parameter estimation to identify the time when the performance drops below a worst-tolerable threshold. Next, we use hypothesis testing to predict whether the performance at a future time will still be acceptable. We validate the effectiveness of our proposed forecasting algorithms via extensive training on various LLM benchmark datasets.

### 4.1 ASSUMPTIONS AND DEFINITIONS

To facilitate the presentation of the proposed performance forecasting algorithms, we begin by mathematically modeling the time series for LLM-patching performance with assumptions justified by the statistical observations in Sec-

Figure 3: Illustration of the idea of the forecasting algorithm for the worst-tolerable performance thresholds, $\rho$ and $\gamma$. Ground-truth model $g(t) = \beta_0 + \beta_1 t$ is shown by the solid line. The scatter points represent observed performance indicators of the ground-truth model corrupted by noise. The thresholds $\rho$ and $\gamma$ of worst-tolerable performance are described by the dotted lines. The times at which performance drops below these thresholds, $t_\rho^*$ and $t_\gamma^*$, are denoted by dashed lines.

tion 3.3. Figure 3 shows in circles the temporal performance indicators $y_t$ associated with downstream task, with empty circles denoting future data points. We assume per the statistical evidence in Section 3.3 that the data points are observations from a linear statistical model with a regression function/ground-truth model $g(t) = \beta_0 + \beta_1 t$ represented by the solid line, corrupted by additive white Gaussian noise (AWGN) $\epsilon_t$ with variance $\sigma_\epsilon^2$. Here, $\beta_0$ is the intercept denoting the theoretical performance at $t = 0$ and $\beta_1$ is the slope denoting the rate of change of the performance. This linear statistical model captures the overall trend of the gradual performance degradation and the fluctuations caused by various random factors as observed in Section 3.3.

Downstream developers wishing to forecast future performance may have predefined standards for acceptable performance. We formalize these standards as thresholds that are used in formulation of our algorithms. The horizontal dashed lines in Figure 3 represent two ways in which the thresholds may be defined to accommodate different forecasting needs. In the first case of the dashed line located $\rho$ below the theoretical performance $\beta_0$ at $t = 0$, the downstream developer is concerned with relative performance. For example, suppose the downstream developer will tolerate a $\rho = 5\%$ drop from the theoretical performance at $t = 0$. In the second case, the downstream developer has an absolute threshold $\gamma$ as their worst-tolerable performance, e.g., an accuracy of at least $\gamma = 80\%$ is required. Mathematically, the performance cutoff time $t_\rho^*$ and $t_\gamma^*$, determined by the relative and absolute thresholds $\rho$ and $\gamma$, respectively, fulfills the following constraints:

$$\text{(Relative)} \ \beta_0 - g(t_\rho^*) = \rho, \quad \text{(Absolute)} \ g(t_\gamma^*) = \gamma. \tag{1}$$

Figure 3 depicts the steps to obtain $t_\rho^*$ or $t_\gamma^*$ via the ground-truth model $g(t)$. In the main paper, we focus on formulating parameter estimation and hypothesis testing algorithms for the relative thresholding scenario. See Appendix F for further estimation formulation for absolute thresholding.

## 4.2 PARAMETER ESTIMATION FORMULATION

When a worst-tolerable threshold of the patch performance is specified, future planning can be facilitated by forecasting the time at which the PortLLM patch will need to be retrained. We ask:

| **RQ 1** | Given past performance indicators and a worst-tolerable performance threshold, at what future time will patching performance drop below the threshold? |
|---|---|

We denote available time steps as $\{t_0, t_1, \ldots, t_I\}$, where $t_I$ is the current time. We generalize the linear statistical model presented in Section 4.1 to the cases where $n_{t_i} \geq 1$ performance indicators are available at each time step $t_i$, although for most resource-constrained applications, we expect only one observation per time step, i.e., $n_{t_i} = 1$. Formally, we denote the $j$th performance indicator at time $t_i$ as

$$y_{t_i}^{(j)} \sim \mathcal{N}(\beta_0 + \beta_1 t_i, \sigma_\epsilon^2), \quad i \in \{0, 1, \ldots, I\} \text{ and } j \in \{1, \ldots, n_{t_i}\}. \tag{2}$$

We examine a simplified case that, for any fixed $t_i$, the observations $\{y_{t_i}^{(j)}\}_{j=1}^{n_{t_i}}$ are uncorrelated.

Next, we outline the high-level idea for estimating $t_\rho^*$. First, we note that in the thresholding definition (1), the optimal time step $t_\rho^*$ and the rate of the performance degradation $\beta_1$ are deterministically related, i.e., $\rho = \beta_0 - g(t_\rho^*) = \beta_0 - (\beta_0 + \beta_1 t_\rho^*) = -\beta_1 t_\rho^*$. We can then apply the invariance principle of maximum likelihood estimation (MLE) (Devore et al., 2021) to reduce the problem of estimating $t_\rho^*$ to estimating $\beta_1$, followed by applying the deterministic mapping from $\beta_1$ to $t_\rho^*$. It can be shown that the MLE for $\beta_1$, assuming $n_{t_i} = n$ for all $i$, is

$$\hat{\beta}_1 = \frac{1}{\tilde{t} - \bar{t}^2}(-\bar{t}\,\bar{y}_I + r_I), \tag{3}$$

where $\bar{y}_I = \sum_{i=0}^{I} \sum_{j=1}^{n} y_{t_i}^{(j)}/N$, $r_I = \sum_{i=0}^{I} \sum_{j=1}^{n} t_i y_{t_i}^{(j)}/N$, $N = \sum_{i=0}^{I} n_{t_i}$, $\bar{t} = \sum_{i=0}^{I} t_i/(I+1)$, and $\tilde{t} = \sum_{i=0}^{I} t_i^2/(I+1)$. Detailed steps to derive (3) are provided in Appendix C. Applying the invariance principle using the mapping $t_\rho^* = -\rho/\beta_1$, we obtain the MLE for $t_\rho^*$ as follows:

$$\hat{t}_\rho^* = -\rho/\hat{\beta}_1. \tag{4}$$

In the specific case of uniform time steps, i.e., $t_i = i \ \forall i$, the precision $\hat{t}_\rho^*$ is

$$\text{Var}\left(1/\hat{t}_\rho^*\right) = \frac{\text{Var}(\hat{\beta}_1)}{\rho^2} = \frac{12\sigma_\epsilon^2}{n\rho^2 I(I+1)(I+2)}. \tag{5}$$

Here, $\text{Var}(\hat{\beta}_1)$ is provided in (28) of Appendix C. We note that, as $I$ increases, the precision of the cutoff time estimator decreases at the rate of $O(I^{-3})$, implying that the width of the confidence interval for $1/\hat{t}_\rho^*$ shrinks at the rate of $O(I^{-3/2})$. In the case that the time steps are not consecutive integers, the variance also decreasing with increasing observations, but the variance also depends on the spacing of the time steps. Figure 4 plots theoretical curves for the precision of $t_\rho^*$, where sharp turnings are seen between the duration of observation $I = 2$ and $4$. The estimation is also more accurate for larger relative threshold $\rho$. Experimental validation results using this estimation framework are provided in Section 4.4.

### 4.3 HYPOTHESIS TESTING FORMULATION

A downstream developer may have a time of interest, $t = t_{\mathrm{m}}$, for which the judgment of acceptable performance is required. For example, if the base model is updated quarterly, the time step of interest may be $t_{\mathrm{m}} = 8$, or two years after the patch is initially trained. In this case, the test may be performed at $I = 4$, or one year after the initial training, to determine whether retraining will be needed in the next year's budget. Formally, we ask:

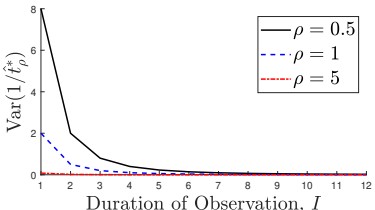

Figure 4: Theoretical plot for the variance of the estimator $1/\hat{t}_\rho^*$ as a function of the duration of observation, $I$. The variance decreases at the rate of $O(I^{-3})$, with sharp turnings between $I = 2$ and $4$. The estimation is more accurate when the duration $I$ is longer and the relative threshold $\rho$ is larger.

| **RQ 2** | Give past performance indicators, a worst-tolerable threshold, and a future time step of interest, will patch performance fail to meet the threshold at the future time step? |
|---|---|

We define the null hypothesis $\mathrm{H}_0$ as the patch fails to meet performance standards at time $t_{\mathrm{m}}$, and the alternative hypothesis $\mathrm{H}_1$, otherwise, namely,

$$\mathrm{H}_0 : \beta_0 - g(t_m) > \rho \quad \text{against} \quad \mathrm{H}_1 : \beta_0 - g(t_m) \leq \rho. \tag{6}$$

This composite hypothesis testing is solved as a likelihood ratio test (LRT) (Hogg et al., 2019) as:

$$\lambda = \frac{\sup_{\boldsymbol{\beta}} L(\boldsymbol{\beta} \in R_0)}{\sup_{\boldsymbol{\beta}} L(\boldsymbol{\beta} \in \mathbb{R}^2)} \overset{\text{def}}{=} \frac{L(\hat{\boldsymbol{\beta}}^{(0)})}{L(\hat{\boldsymbol{\beta}}_{\mathrm{MLE}})} > \eta_0 \quad \text{accepts } \mathrm{H}_0, \tag{7}$$

where $\hat{\boldsymbol{\beta}}^{(0)} = \text{argmax}_{\boldsymbol{\beta} \in R_0} L(\boldsymbol{\beta})$ and $\hat{\boldsymbol{\beta}}_{\mathrm{MLE}} = \text{argmax}_{\boldsymbol{\beta} \in \mathbb{R}^2} L(\boldsymbol{\beta})$. $L(\boldsymbol{\beta})$ defined in (30) of Appendix D is the likelihood of the observations, $R_0$ defined in (29) is the feasible region of $\boldsymbol{\beta}$ for which $\mathrm{H}_0$ is true, and $\eta_0$ is a decision threshold. We show in Appendix D that the test statistic, $z$

$$z = \min(0, e) > \eta \quad \text{accepts } \mathrm{H}_0, \tag{8}$$

where $\eta < 0$ is another decision threshold, $e = -t_m \hat{\beta}_1 - \rho \sim \mathcal{N}(\mu_e, \sigma_e^2)$, $\mu_e \overset{\text{def}}{=} -t_m \beta_1 - \rho$, and $\sigma_e^2 \overset{\text{def}}{=} t_m^2 \text{Var}(\hat{\beta}_1)$. We note that the null hypothesis can also be written as $\mathrm{H}_0 : \mu_e > 0$. Here, $e$ captures the estimated margin by which the patch's performance at $t_m$ exceeds the threshold $\rho$.

The false negative rate (FNR), i.e., the probability of predicting that retraining is not needed at $t_{\mathrm{m}}$ when it is actually needed, and the false positive rate (FPR) are given by

$$\text{FNR} = \mathbb{P}(z < \eta \mid \mu_e > 0) = \Phi(\tfrac{\eta - \mu_e}{\sigma_e}), \quad \text{(9a)}$$

$$\text{FPR} = \mathbb{P}(z > \eta \mid \mu_e \leq 0) = \Phi(\tfrac{\mu_e - \eta}{\sigma_e}). \quad \text{(9b)}$$

Theoretical ROC curves are plotted in Figure 5. We find that while the LRT itself is close to optimal, the hyperparameters control the difficulty of the problems. For example, the test can make more accurate decisions when $\mu_e$ under $\mathrm{H}_0$ and $\mathrm{H}_1$ are more different, or when the duration of observed $I$ is longer. See Section 4.4 for experimental validation for this test.

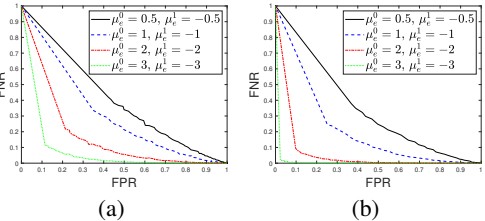

(a)        (b)

Figure 5: Theoretical ROC curves for hypothesis testing. We use time step of interest $t_{\mathrm{m}} = 8$, relative threshold $\rho = 1$, for duration of observations (a) $I = 4$ and (b) $I = 6$. Here, $\mu_e^i$ denotes performance margin for $\mathrm{H}_i$. For $\mu_e^i$ further from zero or longer observation durations, the hypothesis testing tends to be more accurate.

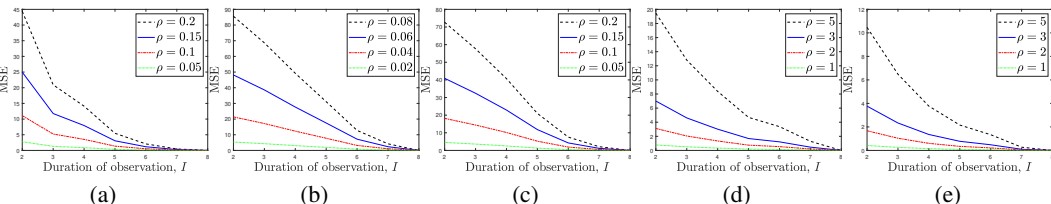

(a)          (b)          (c)          (d)          (e)

Figure 6: MSE of estimate for the time, $\hat{t}_\rho^*$, at which performance drops below the worst-tolerable threshold as a function of the duration of observation, $I$ for different relative thresholds, $\rho$. The curves are averaged across 5 repetitions. The benchmarks are (a) ARC-Easy, (b) BoolQ, (c) WinoGrande, (d) MathGenie (patch #1), and (e) MathGenie (patch #2). MSE decreases with increasing $I$, and the largest drops occur for smaller $I$. As downstream task performance indicators are observed for longer duration, we expect estimates of $\hat{t}_\rho^*$ to be more accurate.

## 4.4 Experimental Validation of the Effectiveness of Proposed Algorithms

We validate our forecasting methods on a base model continually pretrained on UpVoteWeb (UpV, 2024) data and fine-tuned using downstream benchmarks: ARC-Easy (Clark et al., 2018), BoolQ (Clark et al., 2019), WinoGrande (Sakaguchi et al., 2021), and MathGenie (Lu et al., 2024). [3] [4]

**Parameter Estimation.** Figure 7 shows two representative examples of parameter estimation scenarios on repetitions #2 and #3 of ARC-Easy. In Figure 7(a), the upper and lower dashed lines coincide with the predicted performance at $t = 0$, $\hat{\beta}_0$, and the predicted performance minus the threshold, $\hat{\beta}_0 - \rho$. Observed performance drops below the lower dashed line at approximately $t = 5$, while the prediction is at $\hat{t}_\rho^* = 4.3$. In this case, the downstream developer would be correctly informed to retrain before $t = 5$. Similarly, in Figure 7(b), the estimator predicts that retraining is needed slightly before it actually is; in this case, the downstream developer may think that retraining is needed before $t = 4$. For each plot, we used $t_I = I = 3$ so that observations from $t = 0$ to $t = 3$ were used

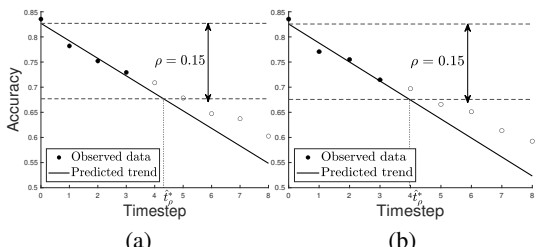

(a)          (b)

Figure 7: Estimation results for tolerance threshold $\rho = 0.15$ on (a) repetitions #2 and (b) #3 of ARC-Easy using $I = 3$ as the latest observation. The estimates are $\hat{t}_\rho^* = 4.31$ for (a) and $\hat{t}_\rho^* = 3.97$ for (b). Both estimates would correctly direct the downstream developer to retrain before $t = 5$.

for estimation. If the base model is updated quarterly, the downstream developer could predict performance after nine months.

To systematically validate our estimation algorithm, we consider the mean squared error (MSE) $(1/M) \sum_{m=1}^M (t_\rho^* - \hat{t}_\rho^*)^2$ where $M = 5$ repetitions and $I$ is the index of the time for the most recently observed data point. From (28), the variance of $\hat{\beta}_1$ decreases at a rate of $O(I^{-3})$. Because $\hat{t}_\rho^*$ is deterministically related to $\hat{\beta}_1$, we expect the MSE of $\hat{t}_\rho^*$ to decrease as $I$ increases. Figure 6 empirically shows that the error decreases when the duration of observation increases. The largest drop in error occurs from $I = 2$ to 3, indicating that estimation will be much more reliable after at least three downstream patching rounds. This drop is especially pronounced for ARC-Easy and MathGenie. For WinoGrande and BoolQ, MSE decreases approximately linearly from $I = 0$ to 6. Figure 8 also reveals that estimation is consistently more accurate for smaller $\rho$ and Figure 16 of Appendix E provides a more direct visualization—when $\rho$ is smaller, $t_\rho^*$ is also smaller, so there is a

---

[3]Because the true parameters $\beta_1$ and $\beta_0$ are unavailable from the real benchmark data, we use the best estimate from 9 available time steps as a proxy for the ground truth.

[4]By observing the data, we found that $t \in [0, 8]$ is the interval for which the trend is best considered linear. This coincides with the practical nature of time of interest for prediction because PortLLM patching performance will be too low to be useful after $t = 8$.

Table 1: Hypothesis testing example on rep. #2 of BoolQ using $\rho = 0.1$ and $t_m = 5$. The test correctly selects $H_1$ when $\eta = -0.001$ but incorrectly decides $H_0$ when $\eta = -0.01$.

| $I$ | Test stat $z$ | Decision thres $\eta = -0.01$ | Decision thres $\eta = -0.001$ |
|---|---|---|---|
| 1 | 0.000 | $H_0$ | $H_0$ |
| 2 | $-0.017$ | $H_1$ | $H_1$ |
| 3 | $-0.002$ | $H_0$ | $H_1$ |
| 4 | $-0.005$ | $H_0$ | $H_1$ |
| 5 | $-0.003$ | $H_0$ | $H_1$ |

Table 2: AUC results for hypothesis testing. Prediction is much more accurate when $\mu_e$ is far from zero and when more time steps have been observed.

| $I$ | $\mu_e^0$ Range | $\mu_e^1$ Range | AUC | Downstream Datasets | Metric |
|---|---|---|---|---|---|
| 5 | $0.05 \pm 0.05$ | $-0.13 \pm 0.03$ | 1.000 | AE, BQ, WG | Acc |
| 5 | $0.50 \pm 0.50$ | $-3.50 \pm 0.50$ | 0.922 | MG | BLEU |
| 5 | $0.05 \pm 0.05$ | $-0.08 \pm 0.03$ | 0.885 | AE, BQ, WG | Acc |
| 5 | $0.50 \pm 0.50$ | $-2.50 \pm 0.50$ | 0.816 | MG | BLEU |
| 4 | $0.05 \pm 0.05$ | $-0.13 \pm 0.03$ | 0.958 | AE, BQ, WG | Acc |
| 4 | $0.50 \pm 0.50$ | $-3.50 \pm 0.50$ | 0.680 | MG | BLEU |
| 4 | $0.05 \pm 0.05$ | $-0.08 \pm 0.03$ | 0.715 | AE, BQ, WG | Acc |
| 4 | $0.50 \pm 0.50$ | $-2.50 \pm 0.50$ | 0.575 | MG | BLEU |

AE: ARC-Easy, BQ: BoolQ, WG: WinoGrande, MG: MathGenie

smaller duration between the time of prediction and the predicted event. Appendix E also presents results on each repetition individually.

Multiple observations per time step ($n \geq 2$) can be obtained by bootstrapping downstream test data with little compute overhead. For example, on ARC-Easy with $\rho = 0.1$, Figure 8 shows that MSE drops substantially from $n = 1$ to $n = 2$, with smaller but noticeable reductions up to $n = 8$. The trend holds across different $\rho$ values, with smaller $\rho$ yielding lower overall error.

**Hypothesis Testing.** Table 1 illustrates our hypothesis testing algorithm using time of interest $t_m = 5$ and the relative threshold $\rho = 0.1$, where the true margin $\mu_e = -0.017 < 0$ falls in the feasible region of $H_1$. This case is difficult since the margin is close to zero. At $I = 3$, the third row of Table 1, the test statistic is $z = -0.002$. For a choice decision threshold $\eta = -0.001$ we decide $H_1$, but for $\eta = -0.01$ we decide $H_0$. A similar pattern holds in the other rows, with correct decisions at $\eta = -0.001$ but typically incorrect at $\eta = -0.01$. This shows the influence of threshold choice on testing when the true margin is near zero, i.e., the decision is inherently a different problem.

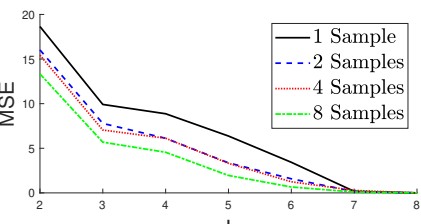

Figure 8: MSE for $\hat{t}_\rho^*$ for $n \geq 1$ observations per time step. The most substantial improvement is observed from $n = 1$ to 2. The results indicate that bootstrapping downstream testing data can increase the prediction accuracy of $\hat{t}_\rho$ for downstream developers.

To further assess general performance, we validate the effectiveness of our hypothesis testing algorithm by calculating area under the curve (AUC) values for ROC curves using experimental data grouped in Table 2. Here, $\mu_e^i$ denotes $\mu_e$ on hypothesis $H_i$. We generate tests by varying the worst-tolerable performance threshold $\rho$ selection and the time step of interest $t_m$ selection across four benchmarks. ROC curves for accuracy tasks (ARC-Easy, BoolQ, WinoGrande) are drawn separately from BLEU tasks (MathGenie). Because the hypotheses can be written as $H_0 : \mu_e > 0$ and $H_1$ otherwise, we group test cases for ranges of $\mu_e$ values. Take the second row as an example, grouping test cases with $\mu_e \in [0, 1]$ for $H_0$ and $\mu_e \in [-4, -3]$ for $H_1$ to calculate error rates and draw the ROC curves yields an AUC of 0.922. Overall, FNR is consistently very low, while the FPR depends on how far $\mu_e$ under $H_1$ lies from zero. Longer duration of observation $I$ also improves test accuracy.

## 5 CONCLUSION

We presented the first systematic study of the temporal performance degradation of patched LLMs. Statistical analysis from large-scale experiments revealed that temporal patching consistently exhibits predictable performance declines as base LLMs evolve. To proactively manage this degradation in deployment, our proposed statistical forecasting tools can advise developers on when and whether patches would fail. Running these tools in simulated PortLLM deployment scenarios demonstrated that estimation precision improves significantly when the observation duration $I$ exceeds 2–4 time steps and when using two performance measures per time step. The proposed statistical forecasting tools can assist downstream developers in making informed technical and business decisions about patch reuse as base LLMs evolve.

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

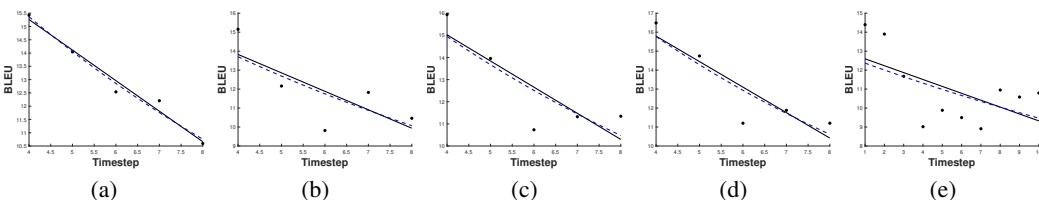

(a)       (b)       (c)       (d)       (e)

Figure 9: Model fitting results using various pretrain data. Linear (solid black lines) and exponential (dashed blue lines) fits are shown for (a) MathGenie repetition 1, (b) MathGenie repetition 3, (c) MathGenie repetition 4, (d) MathGenie repetition 5, and (e) Reddit. As in the case of using the Reddit pretrain dataset, the linear and exponential fits produce similar trends.

## A    DOWNSTREAM FINETUNING DATASETS

MathGenie (Lu et al., 2024) is a synthetic dataset of verified math problems generated by rephrasing and validating questions derived from GSM8K (Cobbe et al., 2021) and MATH (Hendrycks et al., 2021). It supports high-quality evaluation of multi-step mathematical reasoning through natural language explanations.

BoolQ (Clark et al., 2019) is a binary reading comprehension task, consisting of naturally occurring yes/no questions paired with short passages from Wikipedia.

WinoGrande (Sakaguchi et al., 2021) is a large-scale commonsense reasoning dataset structured as pronoun resolution problems. It is designed to be more challenging than the original Winograd Schema Challenge (Levesque et al., 2012), offering high lexical diversity and reduced bias.

ARC-Easy (Clark et al., 2018) is a standardized science Q&A benchmark targeting elementary-school-level multiple-choice questions, emphasizing recalling science facts and basic reasoning.

## B    ADDITIONAL RESULTS ON MODEL FITTING

Results for model fitting where the base model is pretrained on the combined time-series dataset are shown in Figures 9 and 10. It is possible that the temporal variety of pretraining datasets affects the linearity of the trend. Curve fitting plots for pretraining on the Reddit temporal dataset are shown in Figures 11, 12, 13, 14, and 15. The model fitting results support the claim that patch performance on a continuously-updated base model degrades with a predictable trend. In some cases, for example repetitions 2 and 5 in Figures 15 and 12, a linear trend fits the observations well for all time steps. In other cases, the performance degrades linearly for the first part of the time series, then plateaus for later time steps. This case can be observed in repetitions 1 and 3 in Figures 11 and 13. For later time steps, when most of the nonlinearity occurs, the patch performance has degraded below what is acceptable for most applications.

The statistical results for model fitting on linear and exponential models where the base model is pretrained on Reddit data are shown in Table 3. Results for pretraining on various datasets are shown in Table 4. In most cases, there is statistically significant evidence for rejecting a constant trend in favor of a more complex model.

## C    DERIVATION TO OBTAIN CLOSED-FORM SOLUTIONS FOR $\hat{\beta}_0$ AND $\hat{\beta}_1$ AND THEIR VARIANCE

Here, we present the general case where the time steps may not be consecutive integers and there are $n_{t_i} \geq 1$ available observations at each $t_i$. Let the $I$ available time steps be denoted as $\{t_0, t_1, \ldots, t_I\}$ and denote the corresponding $n_{t_i}$ observations at $t_i$ as $\{y_{t_i}^{(1)}, y_{t_i}^{(2)}, \ldots, y_{t_i}^{(n_{t_i})}\}$. We organize all variables for $\hat{\boldsymbol{\beta}}$ estimation into the matrix–vector form below:

Table 3: Curve fitting results for patching an evolved base model continually pretrained on the Reddit temporal dataset. For each test, the null hypothesis ($H_0$) is an intercept model (flat trend). For each repetition and benchmark, we test against linear and exponential trends ($H_1$). For almost all benchmarks, the $p$-values indicate statistically significant evidence against the flat trend. The exponential fit results are obtained by performing a linear fit on log-transformed response variables.

| Benchmark | $H_1$ | $F$-statistic | $p$-value | Adjusted $R^2$ |
|---|---|---|---|---|
| Repetition 1 ARC (Easy) | Linear | 23.9 | $10^{-4}$ | 0.66 |
| Repetition 1 ARC (Easy) | Exponential | 26.0 | $10^{-4}$ | 0.68 |
| Repetition 1 BoolQ | Linear | 10.8 | $10^{-2}$ | 0.45 |
| Repetition 1 BoolQ | Exponential | 10.9 | $10^{-2}$ | 0.45 |
| Repetition 1 MathGenie (patch 1) | Linear | 12.4 | $10^{-3}$ | 0.49 |
| Repetition 1 MathGenie (patch 1) | Exponential | 13.2 | $10^{-3}$ | 0.51 |
| Repetition 1 MathGenie (patch 2) | Linear | 23.8 | $10^{-4}$ | 0.66 |
| Repetition 1 MathGenie (patch 2) | Exponential | 36.9 | $10^{-4}$ | 0.75 |
| Repetition 1 WinoGrande | Linear | 9.0 | $10^{-2}$ | 0.40 |
| Repetition 1 WinoGrande | Exponential | 9.6 | $10^{-2}$ | 0.42 |
| Repetition 2 ARC (Easy) | Linear | 528.6 | $10^{-10}$ | 0.98 |
| Repetition 2 ARC (Easy) | Exponential | 1133.8 | $10^{-12}$ | 0.99 |
| Repetition 2 BoolQ | Linear | 709.2 | $10^{-11}$ | 0.98 |
| Repetition 2 BoolQ | Exponential | 944.3 | $10^{-11}$ | 0.99 |
| Repetition 2 MathGenie (patch 1) | Linear | 118.5 | $10^{-7}$ | 0.91 |
| Repetition 2 MathGenie (patch 1) | Exponential | 206.6 | $10^{-8}$ | 0.94 |
| Repetition 2 MathGenie (patch 2) | Linear | 39.7 | $10^{-4}$ | 0.76 |
| Repetition 2 MathGenie (patch 2) | Exponential | 151.3 | $10^{-7}$ | 0.93 |
| Repetition 2 WinoGrande | Linear | 18.4 | $10^{-3}$ | 0.59 |
| Repetition 2 WinoGrande | Exponential | 20.8 | $10^{-3}$ | 0.62 |
| Repetition 3 ARC (Easy) | Linear | 452.4 | $10^{-10}$ | 0.97 |
| Repetition 3 ARC (Easy) | Exponential | 1102.7 | $10^{-12}$ | 0.99 |
| Repetition 3 BoolQ | Linear | 6.6 | $10^{-2}$ | 0.32 |
| Repetition 3 BoolQ | Exponential | 4.2 | $10^{-1}$ | 0.21 |
| Repetition 3 MathGenie (patch 1) | Linear | 347.9 | $10^{-9}$ | 0.97 |
| Repetition 3 MathGenie (patch 1) | Exponential | 302.9 | $10^{-9}$ | 0.96 |
| Repetition 3 MathGenie (patch 2) | Linear | 34.4 | $10^{-4}$ | 0.74 |
| Repetition 3 MathGenie (patch 2) | Exponential | 150.0 | $10^{-7}$ | 0.93 |
| Repetition 3 WinoGrande | Linear | 23.4 | $10^{-3}$ | 0.65 |
| Repetition 3 WinoGrande | Exponential | 23.4 | $10^{-3}$ | 0.65 |
| Repetition 4 ARC (Easy) | Linear | 213.2 | $10^{-8}$ | 0.95 |
| Repetition 4 ARC (Easy) | Exponential | 261.7 | $10^{-8}$ | 0.96 |
| Repetition 4 BoolQ | Linear | 224.0 | $10^{-8}$ | 0.95 |
| Repetition 4 BoolQ | Exponential | 202.6 | $10^{-8}$ | 0.94 |
| Repetition 4 MathGenie (patch 1) | Linear | 55.8 | $10^{-5}$ | 0.82 |
| Repetition 4 MathGenie (patch 1) | Exponential | 96.8 | $10^{-6}$ | 0.89 |
| Repetition 4 MathGenie (patch 2) | Linear | 44.4 | $10^{-5}$ | 0.78 |
| Repetition 4 MathGenie (patch 2) | Exponential | 97.4 | $10^{-6}$ | 0.89 |
| Repetition 4 WinoGrande | Linear | 33.5 | $10^{-4}$ | 0.73 |
| Repetition 4 WinoGrande | Exponential | 39.2 | $10^{-4}$ | 0.76 |
| Repetition 5 ARC (Easy) | Linear | 321.2 | $10^{-9}$ | 0.96 |
| Repetition 5 ARC (Easy) | Exponential | 335.3 | $10^{-9}$ | 0.97 |
| Repetition 5 BoolQ | Linear | 194.7 | $10^{-8}$ | 0.94 |
| Repetition 5 BoolQ | Exponential | 189.2 | $10^{-8}$ | 0.94 |
| Repetition 5 MathGenie (patch 1) | Linear | 11.2 | $10^{-2}$ | 0.46 |
| Repetition 5 MathGenie (patch 1) | Exponential | 17.0 | $10^{-3}$ | 0.57 |
| Repetition 5 MathGenie (patch 2) | Linear | 25.3 | $10^{-4}$ | 0.67 |
| Repetition 5 MathGenie (patch 2) | Exponential | 24.2 | $10^{-4}$ | 0.66 |
| Repetition 5 WinoGrande | Linear | 30.2 | $10^{-4}$ | 0.71 |
| Repetition 5 WinoGrande | Exponential | 34.0 | $10^{-4}$ | 0.73 |

$$\mathbf{x}_{t_i} = \begin{bmatrix} 1 \\ t_i \end{bmatrix} \in \mathbb{R}^{2 \times 1}, \mathbf{y}_{t_i} = \begin{bmatrix} y_{t_i}^{(1)} \\ \vdots \\ y_{t_i}^{(n_{t_i})} \end{bmatrix} \in \mathbb{R}^{n_t \times 1}, \text{ and } \mathbf{X}_{t_i} = \begin{bmatrix} \mathbf{x}_{t_i}^\top \\ \vdots \\ \mathbf{x}_{t_i}^\top \end{bmatrix} \in \mathbb{R}^{n_{t_i} \times 2}. \tag{10}$$

Table 4: Model fitting statistical results for base models pretrained on the combined time-series dataset. In each case, the alternative hypothesis $H_1$ is tested against an intercept trend. Only 5 time steps from $t = 4$ to $t = 8$ are available for MathGenie. 16 out of 22 cases are statistically significant with $p$-value $< 0.05$ against the flat trend.

| Benchmark | $H_1$ | $F$-statistic | $p$-value | Adjusted $R^2$ |
|---|---|---|---|---|
| Repetition 1 MathGenie | Linear | 107.5 | $10^{-3}$ | 0.96 |
| Repetition 1 MathGenie | Exponential | 106.2 | $10^{-3}$ | 0.96 |
| Repetition 3 MathGenie | Linear | 3.7 | $10^{-1}$ | 0.41 |
| Repetition 3 MathGenie | Exponential | 3.4 | $10^{-1}$ | 0.37 |
| Repetition 4 MathGenie | Linear | 7.4 | $10^{-1}$ | 0.61 |
| Repetition 4 MathGenie | Exponential | 6.9 | $10^{-1}$ | 0.59 |
| Repetition 5 MathGenie | Linear | 11.2 | $10^{-2}$ | 0.72 |
| Repetition 5 MathGenie | Exponential | 10.9 | $10^{-2}$ | 0.71 |
| Reddit | Linear | 4.1 | $10^{-1}$ | 0.25 |
| Reddit | Exponential | 3.3 | $10^{-1}$ | 0.21 |
| Repetition 1 ARC (Easy) | Linear | 12.3 | $10^{-2}$ | 0.62 |
| Repetition 1 ARC (Easy) | Exponential | 12.1 | $10^{-2}$ | 0.61 |
| Repetition 2 ARC (Easy) | Linear | 9.9 | $10^{-2}$ | 0.56 |
| Repetition 2 ARC (Easy) | Exponential | 9.4 | $10^{-2}$ | 0.54 |
| Repetition 1 BoolQ | Linear | 10.6 | $10^{-2}$ | 0.58 |
| Repetition 1 BoolQ | Exponential | 10.6 | $10^{-2}$ | 0.58 |
| Repetition 2 BoolQ | Linear | 92.2 | $10^{-4}$ | 0.93 |
| Repetition 2 BoolQ | Exponential | 93.2 | $10^{-4}$ | 0.93 |
| Repetition 1 WinoGrande | Linear | 6.5 | $10^{-2}$ | 0.44 |
| Repetition 1 WinoGrande | Exponential | 6.2 | $10^{-2}$ | 0.43 |
| Repetition 2 WinoGrande | Linear | 143.6 | $10^{-5}$ | 0.95 |
| Repetition 2 WinoGrande | Exponential | 127.0 | $10^{-5}$ | 0.95 |

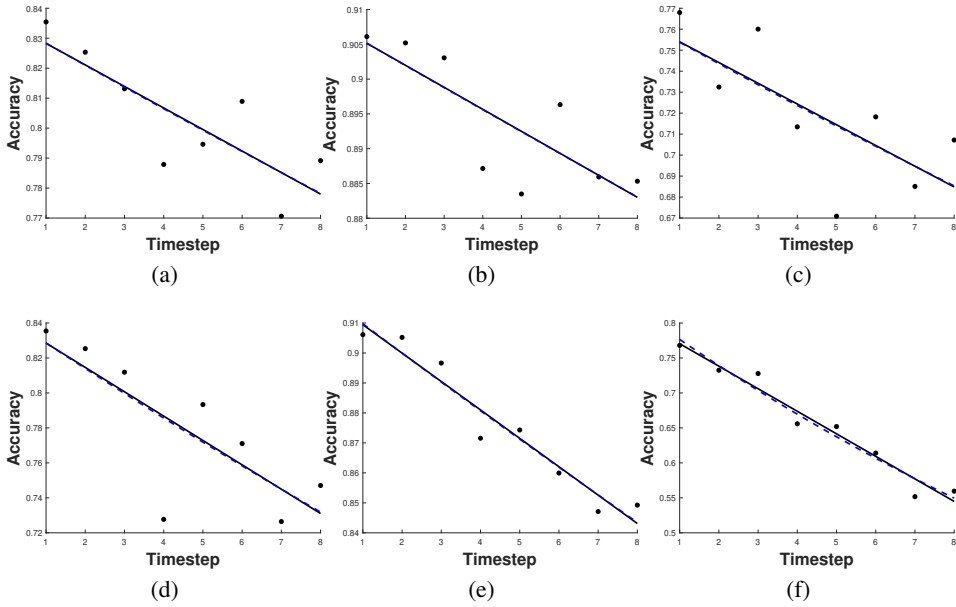

Figure 10: Curve fitting results for linear fit (solid black line) and exponential fit (dashed blue line) for models pretrained on the time series collection dataset. The two rows show two repetitions, and the benchmarks are (a) and (d) ARC-Easy, (b) and (e) BoolQ, and (c) and (f) WinoGrande. As for the evaluations where the Reddit pretrain dataset is used, the linear and exponential curves predict a similar trend.

Further, we define

$$\tilde{\mathbf{X}}_T = \begin{bmatrix} \mathbf{X}_{t_0} \\ \mathbf{X}_{t_1} \\ \vdots \\ \mathbf{X}_{t_I} \end{bmatrix} \in \mathbb{R}^{N \times 2} \text{ and } \tilde{\mathbf{y}}_T = \begin{bmatrix} \mathbf{y}_{t_0} \\ \mathbf{y}_{t_1} \\ \vdots \\ \mathbf{y}_{t_I} \end{bmatrix} \in \mathbb{R}^{N \times 1} \tag{11}$$

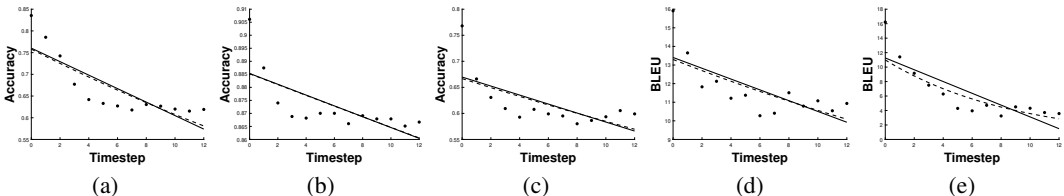

Figure 11: Curve fitting plots for patching an evolved base model pretrained on the Reddit temporal dataset (repetition 1). Benchmarks are (a) ARC-Easy, (b) BoolQ, (c) WinoGrande, (d) MathGenie's patch 1 and (e) MathGenie's patch 2.

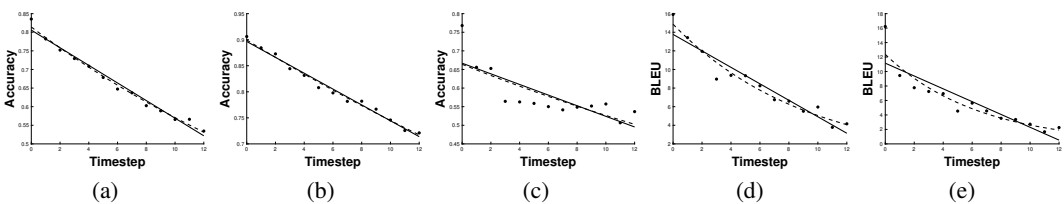

Figure 12: Curve fitting plots for patching an evolved base model pretrained on the Reddit temporal dataset (repetition 2). Benchmarks are (a) ARC-Easy, (b) BoolQ, (c) WinoGrande, (d) MathGenie's patch 1 and (e) MathGenie's patch 2.

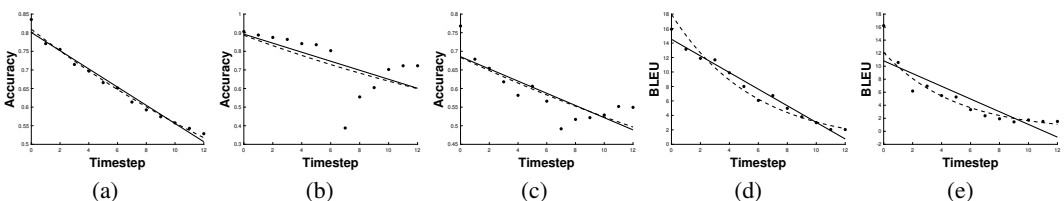

Figure 13: Curve fitting plots for patching an evolved base model pretrained on the Reddit temporal dataset (repetition 3). Benchmarks are (a) ARC-Easy, (b) BoolQ, (c) WinoGrande, (d) MathGenie's patch 1 and (e) MathGenie's patch 2.

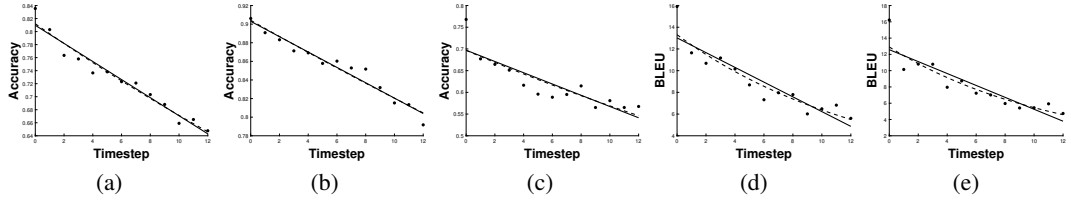

Figure 14: Curve fitting plots for patching an evolved base model pretrained on the Reddit temporal dataset (repetition 4). Benchmarks are (a) ARC-Easy, (b) BoolQ, (c) WinoGrande, (d) MathGenie's patch 1 and (e) MathGenie's patch 2.

where $N = \sum_{t=1}^{I} n_{t_i}$. With these definitions, we can write the maximum likelihood solution,

$$\hat{\boldsymbol{\beta}}_{\text{MLE}} = (\tilde{\mathbf{X}}_T^\top \tilde{\mathbf{X}}_T)^{-1} \tilde{\mathbf{X}}_T^\top \tilde{\mathbf{y}}_T. \tag{12}$$

Taking a special case of $n_{t_i} = n, \forall i \in \{0, 1, \ldots, I\}$, we can develop a closed-form solution for $\hat{\boldsymbol{\beta}}$. In this case, $\tilde{\mathbf{X}}_T^\top \tilde{\mathbf{X}}_T$ can be written as

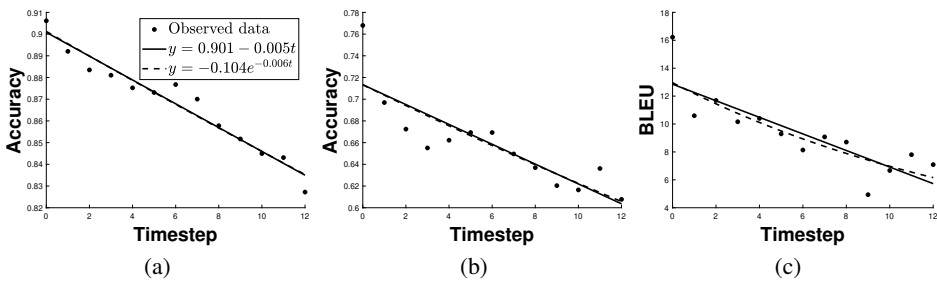

Figure 15: Curve fitting plots for patching an evolved base model pretrained on the Reddit temporal dataset (repetition 5). Benchmarks are (a) BoolQ, (b) WinoGrande, and (c) MathGenie's patch 2.

$$\tilde{\mathbf{X}}_T^\top \tilde{\mathbf{X}}_T = \begin{bmatrix} \sum_{i=0}^{I} \sum_{j=1}^{n} 1 & \sum_{i=0}^{I} \sum_{j=1}^{n} t_i \\ \sum_{i=0}^{I} \sum_{j=1}^{n} t_i & \sum_{i=0}^{I} \sum_{j=1}^{n} t_i^2 \end{bmatrix} = n(I+1) \begin{bmatrix} 1 & \bar{t} \\ \bar{t} & \tilde{t} \end{bmatrix}, \tag{13}$$

where

$$\bar{t} = \frac{1}{I+1} \sum_{i=0}^{I} t_i \quad \text{and} \quad \tilde{t} = \frac{1}{I+1} \sum_{i=0}^{I} t_i^2. \tag{14}$$

The inverse is given by

$$(\tilde{\mathbf{X}}_T^\top \tilde{\mathbf{X}}_T)^{-1} = \frac{1}{n(I+1)(\tilde{t} - \bar{t}^2)} \begin{bmatrix} \tilde{t} & -\bar{t} \\ -\bar{t} & 1 \end{bmatrix}. \tag{15}$$

Now,

$$\tilde{\mathbf{X}}_T^\top \tilde{\mathbf{y}}_T = n(I+1) \begin{bmatrix} \bar{y}_I & r_I \end{bmatrix}^\top \tag{16}$$

where

$$\bar{y}_I \stackrel{\text{def}}{=} \frac{1}{N} \sum_{i=0}^{I} \sum_{j=1}^{n} y_{t_i}^{(j)} \text{ and} \tag{17a}$$

$$r_I \stackrel{\text{def}}{=} \frac{1}{N} \sum_{i=0}^{I} \sum_{j=1}^{n} t_i y_{t_i}^{(j)}. \tag{17b}$$

Hence, (12) can be simplified to

$$\hat{\boldsymbol{\beta}} = \frac{1}{\tilde{t} - \bar{t}^2} \begin{bmatrix} \tilde{t}\,\bar{y}_I - \bar{t}\,r_I \\ -\bar{t}\,\bar{y}_I + r_I \end{bmatrix}. \tag{18}$$

Note that in the case $t_i = i \ \forall i$, $\bar{t} = (I+1)^{-1} \sum_i i = I/2$ and $\tilde{t} = (I+1)^{-1} \sum_i i^2 = I(2I+1)/6$. In this case, (18) simplifies to

$$\hat{\boldsymbol{\beta}} = \frac{2}{I(I+2)} \begin{bmatrix} I(2I+1)\bar{y}_I - 3Ir_I \\ -3I\bar{y}_I + 6r_I \end{bmatrix}. \tag{19}$$

With the solution for $\hat{\boldsymbol{\beta}}$ formulated, it is instructive to also consider the variance of the estimator. The variances and covariance for the statistics are

$$\text{Var}(\bar{y}_I) = \frac{\sigma_\epsilon^2}{N} = \frac{\sigma_\epsilon^2}{n(I+1)}, \tag{20}$$

$$\text{Var}(r_I) = \text{Var}\Big( \frac{1}{N} \sum_{i=0}^{I} \sum_{j=1}^{n} t_i y_{t_i}^{(j)} \Big) = \frac{1}{N^2} \sum_{i=0}^{I} \sum_{j=1}^{n} \text{Var}(t_i y_{t_i}^{(j)}) \tag{21}$$

$$= \frac{\sigma_\epsilon^2 n}{n^2(I+1)^2} \sum_{i=0}^{I} t_i^2 = \frac{\sigma_\epsilon^2 \tilde{t}}{n(I+1)} = \tilde{t} \text{Var}(\bar{y}_I) \tag{22}$$

and

$$\text{Cov}(\bar{y}_I, r_I) = \frac{1}{n^2(I+1)^2} \sum_{i=0}^{I} \sum_{j=1}^{n} t_i \mathbb{E}[(y_{t_i}^{(j)} - \mathbb{E}[y_{t_i}^{(j)}])^2] = \frac{1}{n^2(I+1)^2} \sum_{i=0}^{I} \sum_{j=1}^{n} t_i \sigma_\epsilon^2 \tag{23}$$

$$= \frac{\sigma_\epsilon^2 n}{n^2(I+1)^2} (I+1)\bar{t} = \frac{\sigma_\epsilon^2 \bar{t}}{n(I+1)} = \bar{t} \text{Var}(\bar{y}_I). \tag{24}$$

Note that in the last two equations, we use the independence of the $y_{t_i}^{(j)}$ and $y_{t'_i}^{(j')}$ observations for all $(i,j) \neq (i',j')$ to move the variance operator inside of the summation.

The variance of $\hat{\beta}_0$ in (18) is given by

$$\text{Var}(\hat{\beta}_0) = \frac{1}{(\tilde{t} - \bar{t}^2)^2} [\tilde{t}^2 \text{Var}(\bar{y}_I) + \bar{t}^2 Var(r_I) - 2\bar{t}\tilde{t}\text{Cov}(\bar{y}_I, r_I)] \tag{25a}$$

$$= \frac{1}{(\tilde{t} - \bar{t}^2)^2} \left[ \tilde{t}^2 \frac{\sigma_\epsilon^2}{n(I+1)} + \tilde{t}\bar{t}^2 \frac{\sigma_\epsilon^2}{n(I+1)} - 2\tilde{t}\bar{t}^2 \frac{\sigma_\epsilon^2}{n(I+1)} \right] \tag{25b}$$

$$= \frac{\sigma_\epsilon^2 \tilde{t}}{n(I+1)(\tilde{t} - \bar{t}^2)^2} (\tilde{t} - \bar{t}^2) \tag{25c}$$

$$= \frac{\tilde{t}}{\tilde{t} - \bar{t}^2} \text{Var}(\bar{y}_I). \tag{25d}$$

The variance of $\hat{\beta}_1$ in (18) is given by

$$\text{Var}(\hat{\beta}_1) = \frac{1}{(\tilde{t} - \bar{t}^2)^2} [\bar{t}^2 \text{Var}(\bar{y}_I) + Var(r_I) - 2\bar{t}\text{Cov}(\bar{y}_I, r_I)] \tag{26a}$$

$$= \frac{1}{(\tilde{t} - \bar{t}^2)^2} \left[ \bar{t}^2 \frac{\sigma_\epsilon^2}{n(I+1)} + \tilde{t} \frac{\sigma_\epsilon^2}{n(I+1)} - 2\bar{t}^2 \frac{\sigma_\epsilon^2}{n(I+1)} \right] \tag{26b}$$

$$= \frac{\sigma_\epsilon^2}{n(I+1)(\tilde{t} - \bar{t}^2)^2} (\tilde{t} - \bar{t}^2) \tag{26c}$$

$$= \frac{1}{\tilde{t} - \bar{t}^2} \text{Var}(\bar{y}_I). \tag{26d}$$

In the case $t_i = i \ \forall i$, $\tilde{t} - \bar{t}^2 = I(I+2)/12$ and

$$\text{Var}(\hat{\beta}_0) = \frac{\tilde{t}}{\tilde{t} - \bar{t}^2} \text{Var}(\bar{y}_I) = \frac{2(2I+1)}{(I+2)} \text{Var}(\bar{y}_I) = \frac{2\sigma_\epsilon^2(2I+1)}{n(I+1)(I+2)}. \tag{27}$$

Similarly,

$$\text{Var}(\hat{\beta}_1) = \frac{1}{\tilde{t} - \bar{t}^2} \text{Var}(\bar{y}_I) = \frac{12}{I(I+2)} \text{Var}(\bar{y}_I) = \frac{12\sigma_\epsilon^2}{nI(I+1)(I+2)}. \tag{28}$$

From (27) and (28), it can be seen that, as $I$ increases, the variance of $\hat{\beta}_0$ goes to zero at the rate of $O(I^{-1})$, and the variance of $\hat{\beta}_1$ goes to zero much more quickly at the rate of $O(I^{-3})$. We expect estimation accuracy for the rate of degradation, $\beta_1$, to improve two orders of magnitude faster than for the theoretical initial performance, $\beta_0$, as more indicators $y_{t_i}^{(j)}$ are observed along the time. Similar trends hold when $t_i \neq i$ in general, but the variance also depends on the spacing of the time steps.

## D FORMULATION OF TEST STATISTIC FOR HYPOTHESIS TESTING

In this section, we use a likelihood ratio to define a test statistic that may be used for hypothesis testing. We begin by defining regions of $\boldsymbol{\beta}$ corresponding to each hypothesis. In the case of the absolute threshold we have

$$R_0 = \{(\beta_0, \beta_1) : \beta_0 - g(t_m) > \rho\} \tag{29}$$

where $\boldsymbol{\beta} \in R_0$ when the null hypothesis is true and $\boldsymbol{\beta} \in R_1 = \mathbb{R}^2 \setminus R_0$ when the alternative hypothesis is true. To proceed with hypothesis testing, define the likelihood of the observed performance parameterized by $\boldsymbol{\beta}$ as

$$L(\boldsymbol{\beta}) = p(\tilde{\mathbf{y}}_T; \boldsymbol{\beta}) \tag{30}$$

and the likelihood ratio test

$$\lambda = \frac{\sup\limits_{\boldsymbol{\beta}:\mathrm{H}_0} L(\boldsymbol{\beta})}{\sup\limits_{\boldsymbol{\beta}} L(\boldsymbol{\beta})} \overset{\text{def}}{=} \frac{L(\hat{\boldsymbol{\beta}}^{(0)})}{L(\hat{\boldsymbol{\beta}}_{\mathrm{MLE}})}, \quad \text{where } \hat{\boldsymbol{\beta}}^{(0)} = \underset{\boldsymbol{\beta} \in R_0}{\mathrm{argmax}} L(\boldsymbol{\beta}) \text{ and } \hat{\boldsymbol{\beta}}_{\mathrm{MLE}} = \underset{\boldsymbol{\beta} \in \mathbb{R}^2}{\mathrm{argmax}} L(\boldsymbol{\beta}) \tag{31}$$

as the ratio of the likelihood parameterized by the best parameters given $\mathrm{H}_0$ to the global maximum of the likelihood function.

The estimate $\hat{\boldsymbol{\beta}}^{(0)}$ is the results of a constrained optimization problems. For this problem, the constraint may be active or inactive. If it is active, the constraint holds with equality. If it is inactive, the optimization result is equivalent to the unconstrained MLE estimation given in (12). In our case, the constraint (with equality) can be written as $\begin{bmatrix} 0 & -t_m \end{bmatrix} \begin{bmatrix} \beta_0 & \beta_1 \end{bmatrix}^\top = \rho$. Defining,

$$\mathbf{c} \overset{\text{def}}{=} \begin{bmatrix} 0 & -t_m \end{bmatrix}^\top \tag{32}$$

and

$$r \overset{\text{def}}{=} \rho \tag{33}$$

we can write the constraint in terms of the known constants, $\mathbf{c}$ and $r$, as

$$\mathbf{c}^\top \boldsymbol{\beta} = r, \quad \mathbf{c} \in \mathbb{R}^{2 \times 1}, r \in \mathbb{R}. \tag{34}$$

Defining $\hat{\boldsymbol{\beta}}_u = \hat{\boldsymbol{\beta}}_{\mathrm{MLE}}$ as the unconstrained results given in (12), it can be shown that

$$\hat{\boldsymbol{\beta}}_c = \hat{\boldsymbol{\beta}}_u - e[\mathbf{c}^\top (\tilde{\mathbf{X}}_T^\top \tilde{\mathbf{X}}_T)^{-1} \mathbf{c}]^{-1} (\tilde{\mathbf{X}}_T^\top \tilde{\mathbf{X}}_T)^{-1} \mathbf{c}, \quad \text{where} \tag{35a}$$

$$e \overset{\text{def}}{=} \mathbf{c}^\top \hat{\boldsymbol{\beta}}_u - r. \tag{35b}$$

This can be viewed as the unconstrained result, $\hat{\boldsymbol{\beta}}_u$, plus a correction term dependent on the error between the unconstrained result and the constraint, $e$. Because the constraint is $\mathbf{c}^\top \boldsymbol{\beta} \geq r$, the constraint is active when $e = \mathbf{c}^\top \boldsymbol{\beta} - r \leq 0$ and if $e \geq 0$ we use unconstrained result $\hat{\boldsymbol{\beta}}_u$ and we wish for the correction term to be zero. Therefore we can consolidate both cases into one equations as

$$\hat{\boldsymbol{\beta}} = \hat{\boldsymbol{\beta}}_u - \min(0, e)[\mathbf{c}^\top (\tilde{\mathbf{X}}_T^\top \tilde{\mathbf{X}}_T)^{-1} \mathbf{c}]^{-1} (\tilde{\mathbf{X}}_T^\top \tilde{\mathbf{X}}_T)^{-1} \mathbf{c}. \tag{36}$$

With the constrained regression problem solved, we turn to finding a test statistic based on the likelihood ratio defined in (31). The log-likelihood ratio is given by

$$\ln \lambda = -\frac{1}{2\sigma_\epsilon^2} \sum_{i=0}^{I} \sum_{j=1}^{n} \left[ (\mathbf{x}_{t_i}^\top \hat{\boldsymbol{\beta}}^{(0)} - y_{t_i}^{(j)})^2 - (\mathbf{x}_{t_i}^\top \hat{\boldsymbol{\beta}}_u - y_{t_i}^{(j)})^2 \right] \tag{37}$$

where we use $\mathbf{x}_{t_i} = [1, t_i]^\top$ from (10). Expanding the squared terms in (37),

$$2\sigma_\epsilon^2 \ln \lambda = \left( \sum_{i=0}^{I} \sum_{j=1}^{n} 2\mathbf{x}_{t_i}^\top y_{t_i}^{(j)} \right)(\hat{\boldsymbol{\beta}}^{(0)} - \hat{\boldsymbol{\beta}}_u) + \sum_{i=0}^{I} \sum_{j=1}^{n} (\mathbf{x}_{t_i}^\top \hat{\boldsymbol{\beta}}_u)^2 - \sum_{i=0}^{I} \sum_{j=1}^{n} (\mathbf{x}_{t_i}^\top \hat{\boldsymbol{\beta}}^{(0)})^2 \tag{38}$$

Note that the left-hand side is an increasing function of $\lambda$. From (36),

$$\hat{\boldsymbol{\beta}}^{(0)} - \hat{\boldsymbol{\beta}}_u = -\min(e, 0)[\mathbf{c}^\top (\tilde{\mathbf{X}}_T^\top \tilde{\mathbf{X}}_T)^{-1} \mathbf{c}]^{-1} (\tilde{\mathbf{X}}_T^\top \tilde{\mathbf{X}}_T)^{-1} \mathbf{c}. \tag{39}$$

Further, note that $\sum_{i=0}^{I} \sum_{j=1}^{n} 2\mathbf{x}_t^\top y_{t_i}^{(j)} = 2\tilde{\mathbf{y}}_T^\top \tilde{\mathbf{X}}_T$, so the first term of (38) can be written as

$$\left(\hat{\boldsymbol{\beta}}^{(0)} - \hat{\boldsymbol{\beta}}_{\text{MLE}}\right)\left(\sum_{i=0}^{I}\sum_{j=1}^{n} 2\mathbf{x}_t^\top y_{t_i}^{(j)}\right) = -2\min(e,0)[\mathbf{c}^\top(\tilde{\mathbf{X}}_T^\top\tilde{\mathbf{X}}_T)^{-1}\mathbf{c}]^{-1}\tilde{\mathbf{y}}_T^\top\tilde{\mathbf{X}}_T(\tilde{\mathbf{X}}_T^\top\tilde{\mathbf{X}}_T)^{-1}\mathbf{c}$$

$$(40\text{a})$$

$$= -2\min(e,0)[\mathbf{c}^\top(\tilde{\mathbf{X}}_T^\top\tilde{\mathbf{X}}_T)^{-1}\mathbf{c}]^{-1}\hat{\boldsymbol{\beta}}_u^\top\mathbf{c} \qquad (40\text{b})$$

The second and third terms in (38) can be written as

$$\sum_{i=0}^{I}\sum_{j=1}^{n}(\mathbf{x}_{t_i}^\top\hat{\boldsymbol{\beta}}_u)^2 = \hat{\boldsymbol{\beta}}_u^\top\tilde{\mathbf{X}}_T^\top\tilde{\mathbf{X}}_T\hat{\boldsymbol{\beta}}_u \text{ and } \sum_{i=0}^{I}\sum_{j=1}^{n}(\mathbf{x}_{t_i}^\top\hat{\boldsymbol{\beta}}^{(0)})^2 = \hat{\boldsymbol{\beta}}^{(0)\top}\tilde{\mathbf{X}}_T^\top\tilde{\mathbf{X}}_T\hat{\boldsymbol{\beta}}^{(0)}. \qquad (41)$$

From (36),

$$\hat{\boldsymbol{\beta}}^{(0)\top}\tilde{\mathbf{X}}_T^\top\tilde{\mathbf{X}}_T\hat{\boldsymbol{\beta}}^{(0)} = \hat{\boldsymbol{\beta}}_u^\top\tilde{\mathbf{X}}_T^\top\tilde{\mathbf{X}}_T\hat{\boldsymbol{\beta}}_u + \min(e,0)[\mathbf{c}^\top(\tilde{\mathbf{X}}_T^\top\tilde{\mathbf{X}}_T)^{-1}\mathbf{c}]^{-1}\mathbf{c})^{-1}[\min(0,e) - 2\hat{\boldsymbol{\beta}}_u^\top\mathbf{c}]. \qquad (42)$$

From (39), (41), and (42), (38) can be written as

$$2\sigma_\epsilon^2\ln\lambda = \min(0,e)[\mathbf{c}^\top(\tilde{\mathbf{X}}_T^\top\tilde{\mathbf{X}}_T)^{-1}\mathbf{c}]^{-1}\left[-2\hat{\boldsymbol{\beta}}_u^\top\mathbf{c} - (\min(0,e) - 2\hat{\boldsymbol{\beta}}_u^\top\mathbf{c})\right] \qquad (43\text{a})$$

$$= -(\min(e,0))^2[\mathbf{c}^\top(\tilde{\mathbf{X}}_T^\top\tilde{\mathbf{X}}_T)^{-1}\mathbf{c}]^{-1} \qquad (43\text{b})$$

From (43), we may define a test statistic

$$z = \min(0,e) \qquad (44)$$

as an increasing function of the likelihood ratio. To better understand how to perform tests on $z$, we consider the distribution of $e$, defined in (35b). Note that, using the definitions of $\mathbf{c}$ and $r$ in (32) and (33),

$$e = -t_m\hat{\beta}_{1,\text{MLE}} - \rho. \qquad (45)$$

The distribution of $e$ is determined by the distribution of $\hat{\beta}_{1,\text{MLE}}$ which is a weighted sum of the (normally distributed) observations, $\{y_{t_i}^{(j)}\}$, so $e$ is also Gaussian. In the case of linear regression where the observations $\{y_{t_i}^{(j)}\}$ are Gaussian and independent, the MLE estimator is equivalent to the least squares (LS) estimator, which is unbiased. Because the MLE estimator is unbiased, the mean of $e$ is given by

$$\mathbb{E}[e] = -t_m\beta_1 - \rho \stackrel{\text{def}}{=} \mu_e \qquad (46)$$

where $\beta_1$ is the deterministic but unknown true model parameter. The variance of $e$ is given by $t_m^2\,\text{Var}(\hat{\beta}_1)$. From (28), when $t_i = i \ \ \forall i$ this is equivalent to

$$\text{Var}(e) = \frac{12t_m^2\sigma_\epsilon^2}{nI(I+1)(I+2)} \stackrel{\text{def}}{=} \sigma_e^2. \qquad (47)$$

where $\sigma_\epsilon^2$ is the noise variance.

# E ADDITIONAL ESTIMATION VALIDATION RESULTS

Here, we show additional estimation validation results. In Figure 16, we plot MSE for $\hat{t}_\rho^*$ against $\rho$. As in Section 4.4, we find that error is smaller for smaller $\rho$.

In Figures 17–26, we plot squared estimation error for $t_\rho^*$ against varying $I$ and $\rho$. These results are the same as in Section 4.4 and Figure 16 except that individual repetitions are shown instead of an average across repetitions.

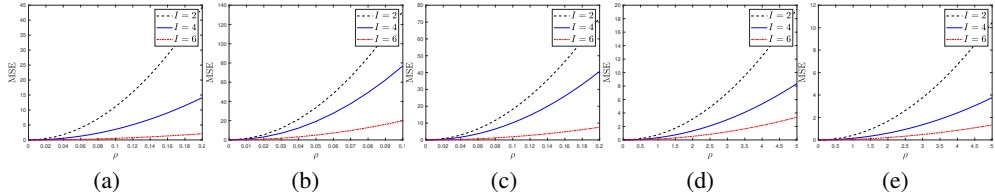

(a)  (b)  (c)  (d)  (e)

Figure 16: MSE of estimate for the time at which performance drops below the worst-tolerable threshold, $\hat{t}^*_\rho$, as a function of the worst-tolerable threshold $\rho$ for maximum available time steps, $I$. The curves are averaged across 5 repetitions. The benchmarks are (a) ARC-Easy, (b) BoolQ, (c), WinoGrande, (d), MathGenie (patch 1), and (e) MathGenie (patch 2). Error increases with increasing $\rho$. Larger $\rho$ requires prediction of time steps further in the future, which makes the estimation task more difficult so that MSE is higher. In general, we expect short-term predictions to be more accurate than long-term predictions.

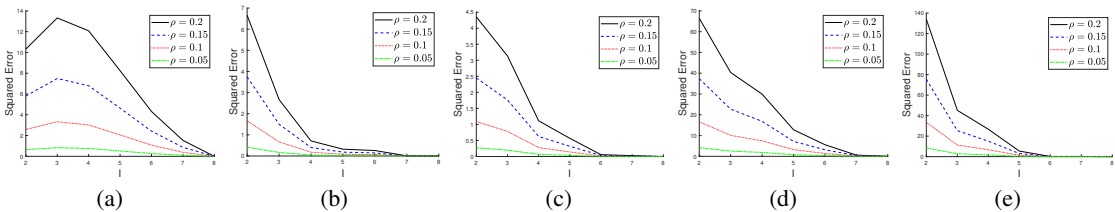

(a)  (b)  (c)  (d)  (e)

Figure 17: Squared error for $t^*_\rho$ estimations on ARC-Easy. Five repetitions are shown. Error is consistently low for $\rho = 0.05$. For larger $\rho$, error depends heavily on $I$. Larger $I$ produces a smaller variance of the estimator.

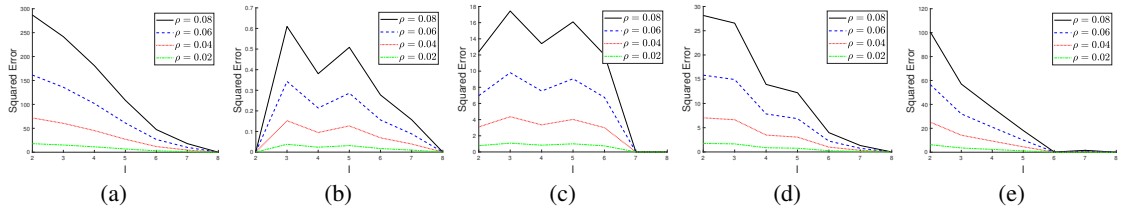

(a)  (b)  (c)  (d)  (e)

Figure 18: Squared error for $t^*_\rho$ estimations on BoolQ. Five repetitions are shown.

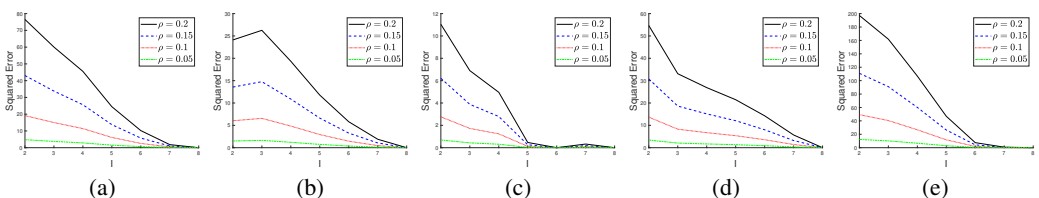

(a)  (b)  (c)  (d)  (e)

Figure 19: Squared error for $t^*_\rho$ estimations on WinoGrande. Five repetitions are shown.

## F    FORMULATIONS FOR THE ABSOLUTE PERFORMANCE THRESHOLD, $\gamma$

Here, we provide formulations for an estimation framework when the worst-tolerable threshold is the absolute threshold, $\gamma$, defined in (1). Here, the time we seek to estimate, $t^*_\gamma$, depends on both the theoretical ground-truth performance at $t = 0$, $\beta_0$, and the slope of performance degradation, $\beta_1$.

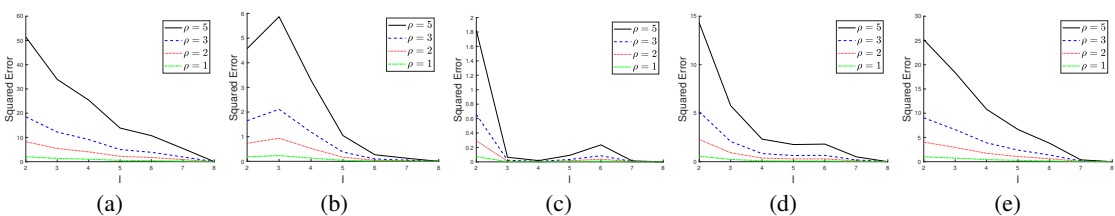

Figure 20: Squared error for $t_\rho^*$ estimations on MathGenie patch 1. Five repetitions are shown.

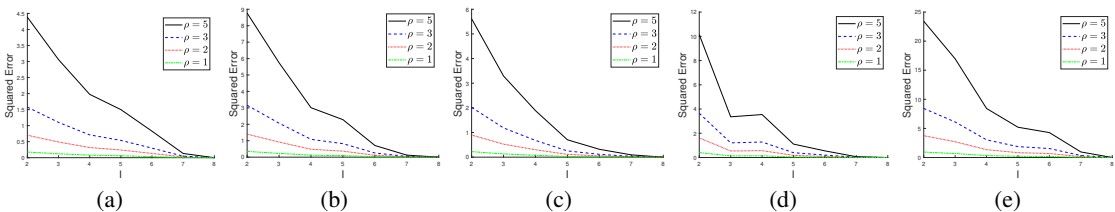

Figure 21: Squared error for $t_\rho^*$ estimations on MathGenie patch 2. Five repetitions are shown.

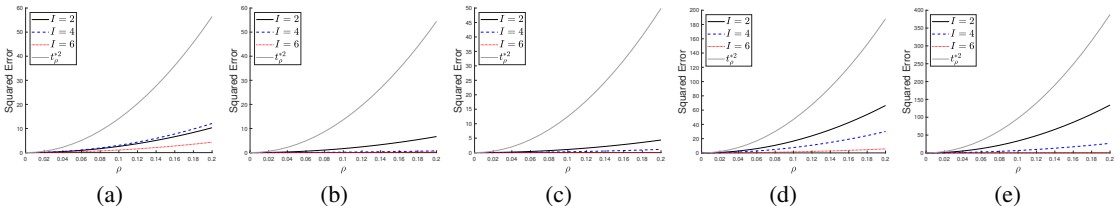

Figure 22: Squared error for $t_\rho^*$ estimations on ARC-Easy. Five repetitions are shown.

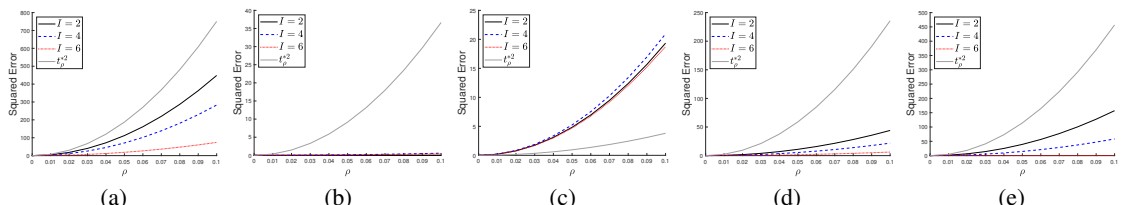

Figure 23: Squared error for $t_\rho^*$ estimations on BoolQ. Five repetitions are shown.

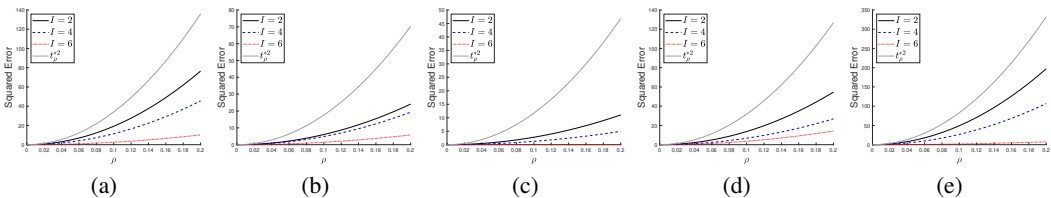

Figure 24: Squared error for $t_\rho^*$ estimations on WinoGrande. Five repetitions are shown.

Namely,

$$t_\gamma^* = g^{-1}(\gamma) = \frac{\gamma - \beta_0}{\beta_1} \tag{48}$$

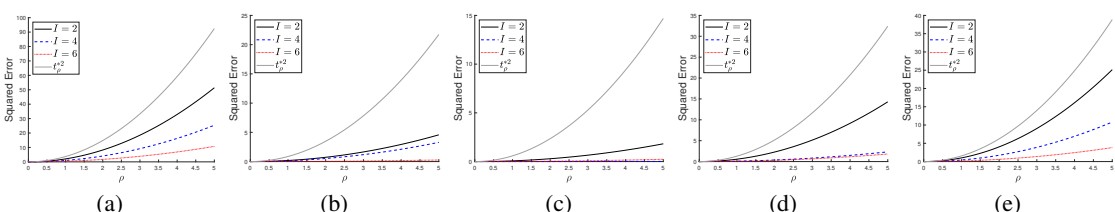

Figure 25: Squared error for $t_\rho^*$ estimations on MathGenie patch 1. Five repetitions are shown.

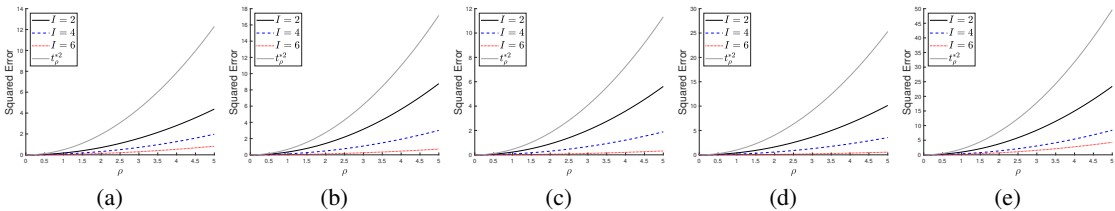

Figure 26: Squared error for $t_\rho^*$ estimations on MathGenie patch 2. Five repetitions are shown.

We begin by showing the estimators for $\beta_0$ and $\beta_1$, then use those estimators to estimate $t_\gamma^*$. It can be shown that the maximum likelihood estimator (MLE) for the theoretical initial performance $\beta_0$ and the rate of performance degradation $\beta_1$, assuming $n_{t_i} = n$ for all $i$, are

$$\begin{bmatrix} \hat{\beta}_0 \\ \hat{\beta}_1 \end{bmatrix} = \frac{1}{\tilde{t} - \bar{t}^2} \begin{bmatrix} \tilde{t}\bar{y}_I - \bar{t}r_I \\ -\bar{t}\bar{y}_I + r_I \end{bmatrix} \tag{49}$$

where $\bar{y}_I = \sum_{i=0}^I \sum_{j=1}^n y_{t_i}^{(j)}/N$, $r_I = \sum_{i=0}^I \sum_{j=1}^n t_i\, y_{t_i}^{(j)}/N$, $N = \sum_{i=0}^I n_{t_i}$, $\bar{t} = \sum_{i=0}^I t_i/(I + 1)$, and $\tilde{t} = \sum_{i=0}^I t_i^2/(I + 1)$. We provide the detailed steps to obtain (49) in Appendix C. By the MLE invariance principle (Devore et al., 2021), we can write the MLE estimator for $t_\gamma^*$ in terms of the MLE estimators for the model parameters, $\beta_1$ and $\beta_0$, as

$$\hat{t}_\gamma^* = (\gamma - \hat{\beta}_0)/\hat{\beta}_1. \tag{50}$$

In Appendix C, we show that,

$$\text{Var}(\hat{\beta}_1) = \frac{12\sigma_\epsilon^2}{nI(I + 1)(I + 2)} \text{ and } \text{Var}(\hat{\beta}_0) = \frac{2\sigma_\epsilon^2(2I + 1)}{n(I + 1)(I + 2)} \tag{51}$$

when $t_i = i \;\; \forall i$. Hence, we expect the variance of $\hat{t}_\gamma^*$ to decrease as $I$ increases.

## G  VARYING PRETRAIN DATASET SIZES

We study the impact of the density of the pretrain datasets on downstream task performance. Theoretical analysis indicates that our algorithms will not be substantially affected by variable dataset size. Consider two datasets from the same temporal period which may be used for the same base model update: $\mathcal{D}_{\text{small}}$ has 20k tokens and $\mathcal{D}_{\text{large}}$ has 200k tokens. Because they are based in the same time period, both datasets will be drawn from the same distribution and are expected to produce the same gradient descent directions. Formally, we write

$$\mathbb{E}[\mathcal{G}(\mathcal{D}_{\text{small}})] = \mathbb{E}[\mathcal{G}(\mathcal{D}_{\text{large}})], \quad \text{Var}(\mathcal{G}(\mathcal{D}_{\text{small}})) > \text{Var}(\mathcal{G}(\mathcal{D}_{\text{large}})). \tag{52}$$

where $\mathcal{G}(\mathcal{D})$ returns the normalized gradient updates from dataset $\mathcal{D}$. This analysis indicates that, while increasing the dataset size reduces the variance of the gradient updates, it does not affect the optimization landscape substantially, hence patch degradation is not substantially affected.

To test this hypothesis, we increased the dataset size from 20 million tokens to 100 million tokens at $t = 3$, $t = 6$, and $t = 9$. We drew each dataset from a 2-week period in UpVoteWeb (UpV, 2024).

Table 5: Downstream scores with base vs. dense datasets at $t = 3, 6, 9$. Differences were not statistically significant ($p > 0.05$ in all cases; $p > 0.1$ in 10/12)

| Task | Timesteps | | | Base/Dense Diff | $p$-Value |
|---|---|---|---|---|---|
| | $t = 3$ | $t = 4$ (base) | $t = 4$ (dense) | | |
| Math Genie | 7.367 | 6.613 | 5.896 | -0.717 | 0.744 |
| BoolQ | 0.856 | 0.85 | 0.869 | 0.019 | 0.254 |
| Arc-Easy | 0.703 | 0.675 | 0.675 | 0.00 | 0.995 |
| WinoGrande | 0.5865 | 0.577 | 0.685 | 0.108 | 0.068 |
| | $t = 6$ | $t = 7$ (base) | $t = 7$ (dense) | | |
| Math Genie | 4.794 | 4.635 | 4.555 | -0.08 | 0.933 |
| BoolQ | 0.834 | 0.824 | 0.845 | 0.021 | 0.435 |
| Arc-Easy | 0.637 | 0.627 | 0.657 | 0.030 | 0.729 |
| WinoGrande | 0.575 | 0.568 | 0.639 | 0.071 | 0.063 |
| | $t = 9$ | $t = 10$ (base) | $t = 10$ (dense) | | |
| Math Genie | 3.938 | 3.518 | 3.375 | -0.143 | 0.847 |
| BoolQ | 0.817 | 0.806 | 0.831 | 0.025 | 0.428 |
| Arc-Easy | 0.607 | 0.592 | 0.624 | 0.032 | 0.554 |
| WinoGrande | 0.556 | 0.572 | 0.610 | 0.038 | 0.195 |

In Table 5, we report the average downstream evaluation results across two repetitions for before training, after training with the base dataset, and after training with the dense dataset. To test the statistical significance of the difference in downstream evaluation results on the two dataset densities, we performed $t$-tests and report $p$-values. We do not observe a statistically significant difference between the base and dense dataset results. The $p$-value is greater than $0.05$ in all cases and greater than $0.1$ in 10 out of 12 cases.

## H  ADDITIONAL BENCHMARKS

Figure 27 presents PortLLM-style patch results on two additional benchmarks: WinoGrande and ARC-Easy. Both tasks display clear degradation trends over time, reinforcing our core finding that patch misalignment increases with continual pretraining of the base model. On WinoGrande, the fixed patch degrades rapidly and eventually underperforms even the unpatched base model. This suggests that for sensitive commonsense reasoning tasks, static patches can be actively detrimental, worsening performance compared to not patching at all. The result highlights a cautionary implication of relying on frozen patches strategies in domains that are especially brittle to representation drift. On ARC-Easy, static patching and periodic patching closely follow the trend of the evolving base model. However, periodic patching performs inconsistently, sometimes performing worse than fixed patching. We attribute this to the limited size of ARC-Easy's training split, which makes it prone to overfitting. Unlike the initial $t_0$ patch, which benefits from a more generalizable optimization trajectory, periodic patches are retrained on narrow snapshots of data without retuning hyperparameters. While tuning could improve periodic patch performance, it would incur significant computational overhead and cause unfair comparison between patches. Together, these results further underscore the practical need for adaptive monitoring and estimation strategies. Reliance on manual refresh intervals or naive reuse of PortLLM-style patches may fail silently or introduce instability in tasks with limited supervision or high sensitivity.

## I  STRUCTURED/COMBINED TIME-SERIES DATASET

To assess patch robustness across varied continual pretraining sources, we performed additional experiments using a collection of independent datasets structured into a time series. Each dataset or subset represents a temporally distinct pretraining phase, simulating the evolving corpora often encountered in production-scale LLM training pipelines. The datasets are ordered as follows:

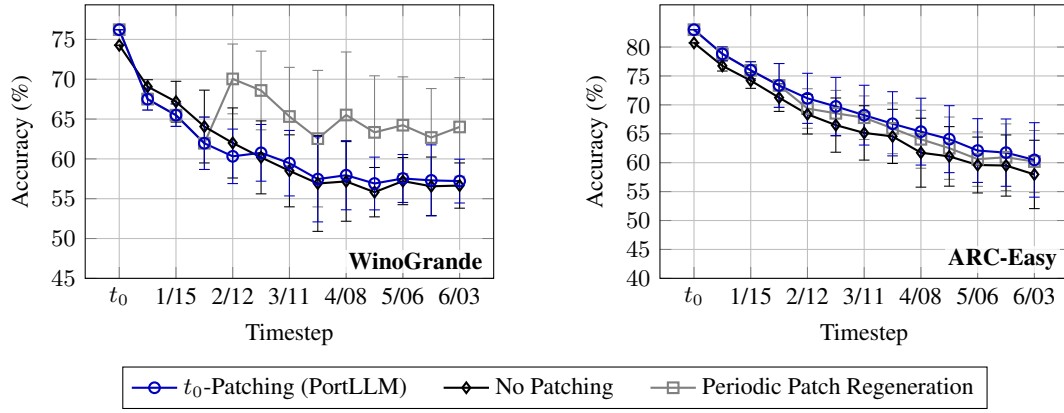

Figure 27: PortLLM patching performance (blue "○") for WinoGrande and ARC-Easy over 13 temporal checkpoints continually pretrained on the Reddit dataset. While PortLLM patching initially outperforms the no-patching baseline (black "◇") as in Khan et al. (2025), our experiments reveal that its performance degrades steadily over time (Section 3). On WinoGrande, periodic patching (gray "□") improves performance at each refresh point. However, for ARC-Easy, periodic patching provides limited gains and occasionally underperforms fixed patching, suggesting sensitivity to task-specific data scarcity or overfitting. Error bars denote the standard deviation across 5 independent repetitions of continually pretrained base models. Forecasting methods (Section 4) may enable more efficient adaptation decisions by anticipating when patch failure occurs.

OpenOrca (Lian et al., 2023) is a large-scale, augmented version of the FLAN Collection (Longpre et al., 2023). It contains approximately 1 million GPT-4 completions and 3.2 million GPT-3.5 completions. OpenPlatypus (Lee et al., 2023) is a reasoning-focused dataset which aggregates multiple logical and scientific reasoning benchmarks such as PRM800K (Lightman et al., 2023), MATH (Hendrycks et al., 2021), ScienceQA (Lu et al., 2022), SciBench (Wang et al., 2024b), ReClor (Yu et al., 2020), and TheoremQA (Chen et al., 2023), applying similarity filtering for question diversity. Math Genie (Lu et al., 2024) is a synthetic dataset of verified math problems generated by rephrasing and validating questions derived from GSM8K (Cobbe et al., 2021) and MATH (Hendrycks et al., 2021). Alpaca-GPT (Peng et al., 2023) is an instruction-following dataset generated by GPT-4 using the original Alpaca prompts (Taori et al., 2023), maintaining the Alpaca format but improving response quality through higher-capability completions. Cosmopedia (Ben Allal et al., 2024) is a synthetic dataset of textbook-style and web-derived documents generated by Mixtral-8x7B-Instruct-v0.1 (Jiang et al., 2024). It contains over 30 million files and 25 billion tokens spanning topics such as math, science, storytelling, and general knowledge. We used three subsets from Cosmopedia: Khan Academy, WikiHow, and OpenStax.

Figure 28 shows patch performance across four benchmarks under temporally structured pretraining sources. These results emphasize the important role of optimization landscape alignment for effective patch transfer. In general, applying a fixed patch from t = 0 leads to consistent degradation over time. However, when the patch and continual pretraining data become aligned, as in MathGenie, performance improves rather than degrades, indicating that landscape compatibility can enhance generalization even without retraining. In contrast, BoolQ, WinoGrande, ARC-Easy exhibit steady performance decay, consistent with our findings in Section 3.2 that attribute degradation to increasing misalignment between the patch and the evolving base model.

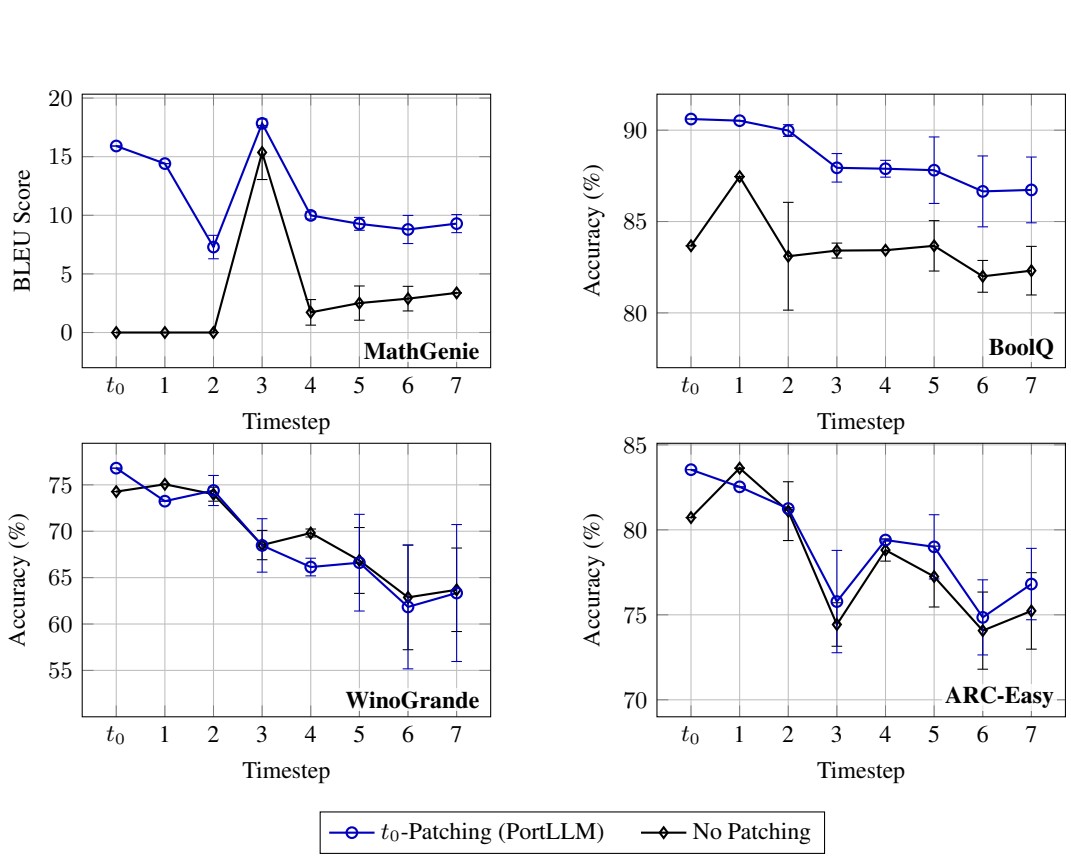

Figure 28: Patch effectiveness across independent pretraining phases structured as a synthetic time series. We evaluate PortLLM-style patch robustness under continual pretraining using 7 distinct datasets: OpenOrca, OpenPlatypus, MathGenie, AlpacaGPT and three subsets from Cosmopedia (Khan Academy, WikiHow and OpenStax). Each dataset represents a temporally isolated pretraining phase, simulating evolving corpora in real-world LLM deployments. As shown, fixed patches trained at $t = 0$ degrade predictably as the base LLM evolves, despite task and architecture remaining constant. But when the patch and base LLM become well-aligned (e.g., MathGenie on MathGenie), patch performance improves over time. Together, these trends highlight the impact of pretraining drift on patch viability and support the hypothesis that degradation arises from misalignment of optimization landscapes.

## J ANALYSIS OF PATCHING PERFORMANCE DEGRADATION ALONG TIME

We mathematically justify that the degradation of the PortLLM-patching performance on a downstream task becomes more severe as $t$ increases.

**Sketch of the proof.** We first show in Lemma 1 that the test error difference between PortLLM-patched model and downstream-fine-tuned model is quadratic in the "distance" between their respective patches. We then need Claim 1 that the patch diverges more with time. We also need Claim 2 for technical conditions.

> **Lemma 1.** PortLLM-patch degradation in terms of test error at time $t$ is quadratic in the difference $\Delta_t$ between the downstream-fine-tuned patches, namely,
>
> $$e(t) \approx \Delta_t^\top \nabla \ell + \Delta_t^\top \nabla^2 \ell \, \Delta_t \tag{53}$$
>
> where $\Delta_t = \phi_0 - \phi_t \in \mathbb{R}^n$ is the difference between the PortLLM patch and the "true" patch trained at time $t$, $\phi_t$ is a vectorized LoRA patch parameter[5] at time $t$, $n$ is the number of LoRA trainable parameters, and $\nabla \ell$ and $\nabla^2 \ell$ are a gradient and Hessian of the testing loss function $\ell$, respectively.

**Claim 1.** The norm of difference $\|\Delta_t\|_2 = \|\phi_0 - \phi_t\|_2$ between the "true" patch $\phi_t$ and PortLLM patch $\phi_0$ increases as a function of $t$. We provide statistical results for this claim in Table 6 of Section J.1. Datasets Arc-Easy and MathGenie provide statistically significant evidence for this claim. See Figure 29 for an intuition in the space of the vectorized expanded patch parameters.

**Claim 2.** The patch is near a local minimum so that the gradient is approximately zero and the Hessian is positive definite, implying that $\Delta_t^\top \nabla \ell \approx 0$ and $\Delta_t^\top \nabla^2 \ell \Delta_t > 0$. These technical conditions are to ensure that $e(t)$ is dominated by the quadratic term. We found that the WinoGrande downstream dataset supports this claim, as shown in Table 7 of Section J.2.

**Proof:** We begin by formally defining the variables and expressions used within our analysis.

Definition 1. Suppose a testing data point/example for a downstream task $(\mathbf{x}, \mathbf{y}) \in \mathcal{X} \times \mathcal{Y}$ is drawn from distribution $\mathcal{D}$ where $\mathcal{X}$ is the set of possible input data points and $\mathcal{Y}$ is the set of possible target labels.

Definition 2. We define the *testing error* as a function of the model parameters $\theta \in \mathbb{R}^N$, where $N$ is the number of model parameters, as follows:

$$L(\theta) = \mathbb{E}[J(g(\mathbf{x}; \theta), \mathbf{y})] \tag{54}$$

where $J : \mathcal{Y} \times \mathcal{Y} \to \mathbb{R}$ is a distance measure, $g : (\mathcal{X} \times \Theta) \to \mathcal{Y}$ represents the inference process of the neural network, and the expectation is taken jointly over the input data $\mathbf{x}$ and target label $\mathbf{y}$.

Definition 3. Patch degradation in terms of test accuracy at time $t$ due to the use of the PortLLM patch may be defined as

$$e(t) = \underbrace{L(\theta_t + \Delta\theta_0)}_{\text{PortLLM patching}} - \underbrace{L(\theta_t + \Delta\theta_t)}_{\text{"True" patching}} \tag{55}$$

for $t = 0, 1, 2, \ldots$ where $\theta_t$ denotes base model parameters at time $t$ and $\Delta\theta_t$ represents a "true"

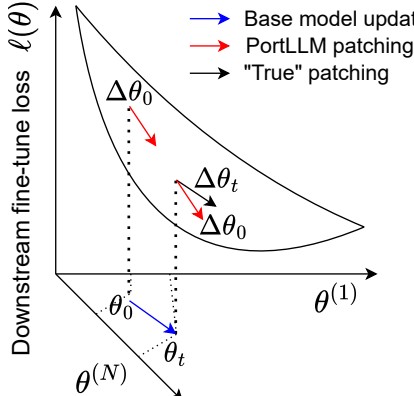

Figure 29: Optimization landscape $\ell(\theta)$ for downstream fine-tuning at time $t$. The "true" descent direction $\Delta\theta_t$ can be calculated from the downstream task, whereas the PortLLM patch's direction $\Delta\theta_0$ exhibits a slightly different direction because it was obtained by downstream-fine-tuning at $\theta_0$. As $t$ increases, we claim that $\Delta\theta_t$ will be increasingly different from $\Delta\theta_0$ as the base model $\theta_t$ evolves. This, in turn, leads to degradation in patch performance in terms of an increased $e(t)$ per (59d).

---

[5]Here, we distinguish between the vectorized expanded patch parameters, $\Delta\theta$, and the vectorized unexpanded patch parameters, $\phi$, as given in (56) and (57) of Definition 5 where $K$ is the number of trainable LoRA matrix pairs. Choosing not to expand the $B_i A_i$ products makes our empirical validations in Table 7 tractable by reducing the number of trainable parameters.

patch downstream-fine-tuned at time $t$ over $\theta_t$. Here, the first term denotes the test error of PortLLM patching and the second term denotes the test error that would be incurred if periodic repatching were performed.

Definition 4. Using the same notation, the performance of the base model directly evaluated on the downstream task without fine-tuning at time $t$ is given by $L(\theta_t)$.

Definition 5. Let $\ell(\phi)$ denote the loss on unexpanded LoRA patch parameters $\phi$, i.e., $\ell(\phi) \overset{\text{def}}{=} L(\theta_t + \Delta\theta)$, where

$$\Delta\theta = \text{vec}(B_1 A_1, \ldots, B_K A_K) \tag{56}$$

is a vector of the expanded LoRA parameters,

$$\phi = \text{vec}(B_1, A_1, \ldots, B_K, A_K) \tag{57}$$

is a vector of the unexpanded LoRA parameters, and $K$ is the number of trainable LoRA matrix pairs.

Main proof. Per Definition 5, we note that $e(t)$ in (55) can be rewritten as

$$e(t) = \ell(\phi_0) - \ell(\phi_t) \tag{58a}$$
$$= \ell(\phi_t + \Delta_t) - \ell(\phi_t) \tag{58b}$$

where $\Delta_t \overset{\text{def}}{=} \phi_0 - \phi_t$. Performing a Taylor series expansion of the first term in (58b) around $\phi_t$, we obtain

$$e(t) = \ell(\phi_t) + \Delta_t^\top \nabla\ell(\phi_t) + \Delta_t^\top \nabla^2\ell(\phi_t)\Delta_t + R_3 - \ell(\phi_t) \tag{59a}$$
$$\approx \Delta_t^\top \nabla\ell(\phi_t) + \Delta_t^\top \nabla^2\ell(\phi_t)\Delta_t \tag{59b}$$
$$\approx \Delta_t^\top \nabla^2\ell(\phi_t)\Delta_t \tag{59c}$$
$$= \|(\nabla^2\ell(\phi_t))^{1/2}\Delta_t\|_2^2 \tag{59d}$$

where $R_3$ represents higher order terms and $\nabla^2\ell(\phi_t)$ denotes the Hessian matrix. (59b) is obtained by assuming a well-behaved $\ell(\cdot)$ such that $R_3$ is small. (59c) is due to Claim 2 on the almost zero gradient. We show that the remaining term is nonnegative by Claim 2 because the Hessian is positive definite. We write it in terms of the Cholesky decomposition of the Hessian to obtain (59d). When the distance $\|\Delta_t\|_2 = \|\phi_0 - \phi_t\|_2$ between the "true" patch $\phi_t$ and PortLLM patch $\phi_0$ increases as a function of $t$ (see Claim 1), we can conclude using (59d) that $e(t)$ increases with $t$. This completes the justification that patch performance degradation on a downstream task becomes more severe as $t$ increases.

### J.1 EMPIRICAL VALIDATION FOR CLAIM 1

To validate Claim 1, we use patches downstream fine-tuned on $4$ datasets at time steps $t \in \{4, 8, 12\}$ across 5 repetitions. We use the following hypothesis test. We consider the trend of $\|\Delta_t\| = \|\phi_0 - \phi_t\|$ as a function of $t$. We test the null hypothesis ($H_0$), that each repetition is modeled by its own intercept model, against the alternative hypothesis ($H_1$), that each repetition is modeled by a linear model with a shared slope across 5 repetitions. Results are shown in Table 6. We find statistically significant evidence that the slope is positive, i.e., that $\|\phi_0 - \phi_t\|$ increases as a function of $t$, for datasets Arc-Easy and MathGenie.

### J.2 EMPIRICAL VALIDATION FOR CLAIM 2

To validate Claim 2, we use the method proposed in Pearlmutter (1994) to efficiently calculate Hessian–vector product of a function $f(\theta)$ as follows

$$\mathbf{Hv} = \lim_{r \to 0} \frac{\nabla f(\boldsymbol{\theta} + r\mathbf{v}) - \nabla f(\boldsymbol{\theta})}{r} = \frac{\partial}{\partial r}\nabla f(\boldsymbol{\theta} + r\mathbf{v})|_{r=0} \tag{60}$$

where $\mathbf{H}$ is the Hessian of $f(\boldsymbol{\theta})$ with respect to $\boldsymbol{\theta}$. Results using this method on WinoGrande data to support Claim 2 are shown in Table 7.[6]

---

[6]In implementation, the patch $\Delta\theta$ is scaled by $\frac{\alpha}{r}$ where $r$ is the LoRA rank and $\alpha$ is the LoRA alpha. In our implementation, this factor is $16/8 = 2$. Scaling $\Delta\theta$ by $k$ is equivalent to scaling $\phi$ by $\sqrt{k}$ because

Table 6: Statistical results for $\|\phi_t\|$ trend. We test $H_0$, an intercept model for each repetition, against $H_1$, a linear model for each repetition with a common slope across repetitions. We report $F$ statistics and $p$-values for each dataset. There is statistically significant evidence of a positive slope for ARC Easy and MathGenie, indicating that $\|\phi_t\|$ increases as a function of $t$.

| Dataset | Estimated slope | $F$ statistic | $p$-value |
|---|---|---|---|
| ARC Easy | 0.015 | 9.77 | 0.02 |
| MathGenie | 0.042 | 6.75 | 0.03 |
| WinoGrande | 0.013 | 0.37 | 0.56 |
| BoolQ | $-0.061$ | 0.02 | 0.89 |

Table 7: Validations for Claim 2 on WinoGrande data. To efficiently calculate Hessian–vector products, we use the method of Pearlmutter (1994) to calculate the linear term, $\Delta_t^\top \nabla \ell(\phi_t)$, and the quadratic term, $\Delta_t^\top \nabla^2 \ell(\phi_t) \Delta_t$ of the Taylor series expansion in (59b). We show two repetition for each time step $t$ and average across 100 downstream test examples. We chose 100 examples to balance statistical significance and computational feasibility. The results show that, on average, the linear terms are negligible whereas the quadratic terms are dominant.

| $t$ | Quadratic term 95% CI | Linear term 95% CI |
|---|---|---|
| 4 | $[3.69, 6.92]$ | $[-0.10, 0.13]$ |
| 4 | $[4.17, 7.92]$ | $[-0.12, 0.17]$ |
| 8 | $[3.03, 5.64]$ | $[-0.02, 0.19]$ |
| 8 | $[4.06, 7.77]$ | $[-0.10, 0.18]$ |
| 12 | $[4.04, 7.06]$ | $[-0.17, 0.14]$ |
| 12 | $[3.98, 7.55]$ | $[-0.16, 0.05]$ |

---

$kBA = (\sqrt{k}B)(\sqrt{k}A)$. To account for this implementation detail, we scale linear terms by $\sqrt{k} = \sqrt{2}$ and quadratic terms by $k = 2$.

