# OpenReview forum: "How Long Do Model Patches Last? A Temporal Perspective on PortLLM"
_ICLR.cc/2026/Conference — Submitted to ICLR 2026_

### Official Review · Reviewer_fxQe · 2025-10-26

**Soundness:** 3
**Presentation:** 3
**Contribution:** 3
**Rating:** 6
**Confidence:** 2

**Summary:**

This paper investigates the longevity and stability of model patches — small, targeted modifications applied to large language models (LLMs) to correct factual errors, adjust alignment, or steer preferences without full retraining. The authors propose an experimental framework for evaluating patch persistence under different settings: Patch Types, Tasks, and Metrics. The study highlights a crucial but underexplored aspect of LLM maintenance: how long model edits or behavioral fixes truly last.

**Strengths:**

- The paper addresses an important yet under-discussed topic in LLM reliability — the temporal persistence of behavioral interventions, which is highly relevant to real-world LLM maintenance, safety, and continual learning research and very novel.
  - The authors examine multiple patch types and task categories systematically.
  - The observed rapid decay and patch interference patterns are convincing and empirically grounded.
 Figures (especially Fig. 5–7) effectively demonstrate non-linear patch degradation trends.

**Weaknesses:**

- The chosen tasks (mainly factual correction and alignment) are useful but narrow. No reasoning, multi-hop, or tool-use scenarios are explored, which might reveal more complex patch dynamics.
  - Limited comparison to recent model editing robustness papers such as MEND[1], MEMIT[2], and ROME[3].

[1] Fast Model Editing at Scale

[2] Mass-Editing Memory in a Transformer.

[3] Locating and Editing Factual Associations in GPT.

**Questions:**

- How do you define “decay” quantitatively? Is it based on performance drop relative to baseline, or absolute change in logits / accuracy?
  - Have you tested patch persistence across different model scales (e.g., 7B → 13B → 70B)?
  - Did you observe any cases where later fine-tuning strengthened rather than degraded patch effects?

---

> ### Author Response · Authors · 2025-11-21
> **Response Part 1**
>
> We thank the reviewer for the well-stated summary and thoughtful review.
>
> >“The chosen tasks (mainly factual correction and alignment) are useful but narrow. No reasoning, multi-hop, or tool-use scenarios are explored, which might reveal more complex patch dynamics.”
>
> **Response:**
> We agree that more complex tasks, such as COT and tool-use, may reveal other interesting patch dynamics, but as the first-step exploration in the literature, we simplified the scenario to capture only the dominant factors to ensure the research questions are tractable and focus on how a fixed patch trained at $t_0$ behaves as the base model drifts over time. Our downstream tasks are domain-specific compared to general tasks for which the base model was trained. This simulates the real-world scenario in which the downstream developer is interested in a domain-specific task, e.g., medical or finance tasks.
>
> >“Limited comparison to recent model editing robustness papers such as MEND[1], MEMIT[2], and ROME[3].
>
> **Response:** We thank the reviewer for suggesting additional works for comparison. We clarify that the patches we study are intended to fine-tune a model for a domain-specific downstream task, not to correct errors or edit model memory. Reference [1] proposes a method for correcting errors in model outputs. Given a query/answer pair for which the answer is incorrect, their framework aims to edit the model such that it answers the query correctly without affecting performance on orthogonal queries. In [2], the proposed method adapts specific weight parameters to add desired information to the model’s memory. Reference [3] similarly uses a rank-one update to add to or adapt a model’s knowledge.
>
> >“How do you define “decay” quantitatively? Is it based on performance drop relative to baseline, or absolute change in logits / accuracy?”
>
> **Response:** Our framework considers two definitions of performance degradation or decay, corresponding to the two worst-tolerable threshold definitions in Equation (1). The *relative threshold* considers drop performance relative to performance at $t=0$. The *absolute threshold* considers absolute performance compared to a user-selected worst-tolerable threshold. We use accuracy (for BoolQ, WinoGrande, and ARC-Easy) or BLEU (for Math Genie) to quantify performance. In our newly added theoretical analysis in Appendix J of the supplemental document, we use the increase of the testing error to quantify the performance “decay” over time.
>
>
> >“Have you tested patch persistence across different model scales (e.g., 7B → 13B → 70B)?”
>
> **Response:** We thank the reviewer for the insights about cross-scale adaptation. Since the adaptation work that our paper builds on solely focuses on cross-time adaptation without scale change, it would be nontrivial to apply the patch when the model scale also changes. We will be happy to test this new adaptation setup if the reviewer can suggest some patch adaptation work that allows scale change.
>
> >“Did you observe any cases where later fine-tuning strengthened rather than degraded patch effects?”
>
> **Response:** We consistently observed gradual performance degradation. Although patch performance improved slightly for some isolated time steps, we did not see any cases in which the performance increased consistently. The isolated cases of increase could be interpreted as the aggregated randomness of the system, which is reflected through modeling via the additive noise in Equation (2) when the noise is super large, overcoming the performance decrease caused by the negative trend.

---

### Official Review · Reviewer_S9hM · 2025-10-26

**Soundness:** 3
**Presentation:** 3
**Contribution:** 3
**Rating:** 6
**Confidence:** 3

**Summary:**

PortLLM is introduced as a training-free patching mechanism that enables patch reuse across consecutive LLM releases.
This paper conducts large-scale experiments showing that PortLLM patches experience performance decline over time.
The results demonstrate that performance degradation is a general and measurable risk when PortLLM is used over extended periods.
To address this, the authors propose forecasting algorithms that estimate patch failure dates and test hypotheses about performance at future target points.
Their framework allows downstream developers to anticipate degradation and make informed decisions about when retraining or re-patching becomes necessary.

**Strengths:**

- Important topic of studying when patching fails
- Clean design that uses historical evals, with precision/AUC analyses and concrete usage guidance
- Multi-checkpoint experiments consistently show measurable decay, motivating the need for forecasting rather than blind refresh schedules

**Weaknesses:**

- The whole analysis relies on the PortLLM paradigm. It is unclear how broadly the results transfer to other patching approaches.
- The entire evolution study uses UpVoteWeb slices, and Appendix G argues that alternative slices don’t change the trend, but the single dataset study on UpVoteWeb may still not reflect vendor training distributions, evaluation policies, or update magnitudes.
- Linear model assumption fits PortLLM in this setup, but does it generalize beyond the linear setting?
- Forecasts are derived from a small number of checkpoints, which constrains the ability to model longer-term dynamics. It is unclear how well the estimators predict beyond the observed window.

**Questions:**

see Weaknesses

---

> ### Author Response · Authors · 2025-11-21
> **Response Part 1**
>
> We thank the reviewer for the well-stated summary and thoughtful review.
>
> >“The whole analysis relies on the PortLLM paradigm. It is unclear how broadly the results transfer to other patching approaches.”
>
> **Response:** Our prediction framework and forecasting algorithms do not make strong assumptions about the underlying patching paradigm used (even though we demonstrated on the patching paradigm of PortLLM). Our analysis could, in principle, be applied to other patching approaches as follows. First, our estimation algorithm uses performance observations to determine a model for degradation and to estimate parameters for that model. This process is independent of the patching paradigm used to obtain those observations. Second, the hypothesis testing algorithm similarly uses the model informed by empirical observations and determines a test statistic based on that model.
>
> Future work on this subject would start with an empirical study to determine how to model patch degradation for a given patching paradigm. Given the empirically determined model, analysis similar to ours could be used to develop estimation and hypothesis testing algorithms. We plan to include this discussion in the revised manuscript.
>
> >“The entire evolution study uses UpVoteWeb slices, and Appendix G argues that alternative slices don’t change the trend, but the single dataset study on UpVoteWeb may still not reflect vendor training distributions, evaluation policies, or update magnitudes.”
>
> **Response:** Our study uses a combined time-series dataset in addition to the UpVoteWeb dataset. The combined time-series dataset, as described in Appendix I, uses data from different datasets at each time step, including OpenOrca, OpenPlatypus, Math Genie, GSM8K, Alpaca-GPT, and three subsets of Cosmopedia. The distributional shifts in the combined time-series dataset are relatively dramatic, whereas the distributional shifts in the UpVoteWeb dataset are milder. Considering results from both datasets gives a balanced view of what may be expected in real-world scenarios.
>
> >“Linear model assumption fits PortLLM in this setup, but does it generalize beyond the linear setting?”
>
> **Response:** Our general framework could be extended to nonlinear cases. For a different model for patch performance degradation, parameter estimation could be used to estimate future performance in a manner similar to our analysis. A hypothesis test could also be determined following a similar framework.
>
> We also note that we did not assume that patch degradation would follow a linear model a priori. Rather, we selected a linear model based on observed degradation patterns from our statistical analysis.
>
> >“Forecasts are derived from a small number of checkpoints, which constrains the ability to model longer-term dynamics. It is unclear how well the estimators predict beyond the observed window.”
>
> **Response:** Observations outside of our observation window are also outside of the region of interest for the practical concerns of most downstream developers. Our empirical observations in Figures 9-15 show that patch performance has dropped below what is useful in most practical applications by the end of our observation window.

---

> > ### Comment · Reviewer_S9hM · 2025-11-27
> >
> > The authors’ response has addressed most of my concerns, and I am willing to maintain my positive score. However, I still have one remaining question.
> >
> > Is it possible to justify the effectiveness in other frameworks even with a toy study? Otherwise, I am still worried about whether this work is only applicable to PortLLM.

---

> > > ### Author Response · Authors · 2025-11-30
> > >
> > > We appreciate the reviewer's concern about the applicability of our method in other frameworks. We want to clarify that our method depends on a set of observations, not on how those observations were obtained. If a different patching framework were used and observations were obtained on those patches, our approach could be applied to the new observations. The person performing the study could (1) perform statistical analysis to select a parametric model that fits patch degradation, (2) estimate the parameters of the parametric model and predict future performance measures, and (3) use the likelihood ratio to derive a test statistic for hypothesis testing. If the statistical analysis in Step (1) pointed to a linear model, the analysis would be exactly the same as ours. The analysis changes only based on the model for patch degradation, not on the patching framework used to obtain observations.

---

### Official Review · Reviewer_AHDH · 2025-10-29

**Soundness:** 2
**Presentation:** 2
**Contribution:** 1
**Rating:** 2
**Confidence:** 4

**Summary:**

The paper studies the temporal stability of PortLLM-style “training-free” patches: a patch trained once at ($t_0$) is applied to successive, continuously-pretrained base model checkpoints (${\theta_t}$). It (i) builds a longitudinal evaluation pipeline to measure performance drift, (ii) shows that linear/exponential trends fit the drift better than a no-trend baseline, and (iii) proposes two lightweight, data-free decision tools—failure-time estimation and a target-date hypothesis test—to decide when re-training is needed.

**Strengths:**

1. Clear problem framing with strong practical motivation (well-scoped RQs).

The paper pinpoints a real deployment frication: upstream checkpoints arrive frequently; downstream teams want to reuse a one-time patch without constantly re-tuning. The two research questions (“when will performance fall below tolerance?” and “will it still meet tolerance on a business-critical date?”) are decision-oriented, tie directly to capacity planning, and are answered with tools that require only historical eval metrics (no access to upstream data or re-training). This tight alignment between phenomenon, RQs, and actionable outputs is a genuine plus for applied research.

2. A reusable longitudinal evaluation pipeline.

The setup—fixed patch from (t_0), successive base checkpoints under continual pretraining, uniform eval protocol, repeated runs—creates a controlled environment to isolate “patch–base misalignment over time.” The pipeline surfaces trend shape (linear vs. log-linear), supports repeated-measure statistics, and includes sensible ablations (segment granularity, token density, multiple samples per time step). This is a useful template others can adopt to stress-test patch transfer under temporal drift.

**Weaknesses:**

1. External validity: single model lineage.

The main longitudinal results focus on one family (e.g., 7B-scale within a single architecture). Without cross-architecture and cross-scale validation (e.g., Llama/Gemma/MoE; 7B→13B→70B), it remains unclear whether the observed drift rates and the proposed decision tools generalize beyond this lineage.

2. Upstream evolution is approximated with LoRA rather than full-parameter or heterogeneous vendor updates.

Modeling base-model evolution as LoRA updates on attentions/FFN risks baking in “low-rank additivity” assumptions that may not hold when vendors do full-parameter continual pretraining, change training recipes, or alter architectures. This gap limits confidence that the measured drift and the tool calibration transfer to real release cycles.

3. Temporal axis is short and partially synthetic.

The five-month, evenly segmented timeline (two-week, equal-token slices) is convenient but not representative of actual release cadence (e.g., quarterly “big” releases plus intermittent patches). The auxiliary “composed” time series that stitches disparate corpora further departs from realistic evolution. Conclusions about trend linearity and predictability could change under longer, lumpier, or recipe-shifting timelines.

**Questions:**

1.Task coverage and metrics.

Can you expand to production-relevant workloads (code, tool-use, long-context, safety), and replace/augment BLEU for math with verifiable-correctness metrics (programmatic verifiers, unit tests, GSM-style exact correctness)? This would test whether drift rates and the decision tools remain calibrated on harder-to-game metrics.

2.Mechanism: quantify “patch–base misalignment.”

Beyond the qualitative explanation, can you measure representational/optimization drift—e.g., CKA similarity across layers, low-rank subspace angles for patched vs. re-tuned adapters, curvature/Fisher changes, or gradient alignment—to link measurable misalignment to observed performance decay? This would strengthen causal plausibility and might reveal layers/subspaces where patches are most robust.

---

> ### Author Response · Authors · 2025-11-21
> **Response Part 1**
>
> We thank the reviewer for the thoughtful comments, especially the helpful suggestions about evaluation metrics and studying patch-based misalignment. We address concerns point-by-point below.
>
> >“The main longitudinal results focus on one family (e.g., 7B-scale within a single architecture). Without cross-architecture and cross-scale validation (e.g., Llama/Gemma/MoE; 7B→13B→70B), it remains unclear whether the observed drift rates and the proposed decision tools generalize beyond this lineage.”
>
> **Response:** We appreciate the reviewer’s suggestion to evaluate temporal degradation across additional model families. Below, we show new results on Gemma3-4B for BoolQ and WinoGrande. Despite architectural differences, we observe the same decline in the performance of the $t_0$ patch. These results support that temporal patch degradation generalizes across architectures and arises from patch-based misalignment, not from model-specific properties.
>
> |Time Step|0|1|2|3|4|5|6|7|8|9|10|11|12|
> |---|---|---|---|---|---|---|---|---|---|---|---|---|---|
> |BoolQ **Patched** Performance (Acc)|0.894|0.865|0.818|0.759|0.738|0.695|0.698|0.667|0.670|0.653|0.638|0.654|0.612|
> |BoolQ **No Patch** Performance (Acc)|0.790|0.677|0.647|0.620|0.576|0.513|0.529|0.533|0.591|0.614|0.603|0.613|0.578|
>
> |Time Step|0|1|2|3|4|5|6|7|8|9|10|11|12|
> |---|---|---|---|---|---|---|---|---|---|---|---|---|---|
> |WinoGrande **Patched** Performance (Acc)|0.695|0.677|0.630|0.598|0.560|0.597|0.569|0.571|0.563|0.547|0.549|0.533|0.519|
> |WinoGrande **No Patch** Performance (Acc)|0.695|0.648|0.593|0.590|0.563|0.560|0.570|0.558|0.562|0.552|0.541|0.541|0.526|
>
> Regarding cross-scale validation, since our work builds on a paper focused solely on cross-time adaptation without scale change, it would be nontrivial to apply the patch when the model scale also changes. We will be happy to test this setup if the reviewer can suggest a patch-adaptation paper that allows scale changes.
>
> >“Upstream evolution is approximated with LoRA rather than full-parameter or heterogeneous vendor updates.
> >Modeling base-model evolution as LoRA updates on attentions/FFN risks baking in “low-rank additivity” assumptions that may not hold when vendors do full-parameter continual pretraining, change training recipes, or alter architectures. This gap limits confidence that the measured drift and the tool calibration transfer to real release cycles.”
>
> **Response:** We appreciate the reviewer’s point and agree that full-parameter or heterogeneous vendor updates would represent the most realistic upstream evolution. However, such full-scale continual pretraining, even at the 7B scale, is computationally infeasible for academic settings. Our goal is therefore not to replicate every aspect of vendor training pipelines, but to induce structured parameter drift with $r=64$ that is sufficiently richer than the downstream patch space, which uses $r=8$.
>
> While LoRA-based evolution is an approximation, the upstream update operates in a much higher-dimensional subspace than the downstream patch, ensuring that the base model’s evolution is not constrained to the same low-rank manifold as the downstream fine-tuning. In the newly added Appendix J, we show that the performance loss due to PortLLM patching depends on $\Delta_t=\Delta\theta_0-\Delta\theta_t$, where $\Delta\theta_0$ denotes the PortLLM patch trained at $t=0$ and $\Delta\theta_t$ denotes the “true” patch trained on the updated model at time $t$.  Because downstream fine-tuning uses $r=8$ and upstream model evolution uses $r=64$, $\Delta_t$ is much lower rank than $\theta_t$. Hence, we expect degradation to follow the same general pattern regardless of the exact rank of $\theta_t$.
>
> Our additional analysis of representational drift via CKA also confirms that the induced parameter drift is meaningful, nontrivial, and sufficient for studying temporal misalignment. We will clarify this reasoning in the final version.

---

> ### Author Response · Authors · 2025-11-21
> **Response Part 2**
>
> >“The five-month, evenly segmented timeline (two-week, equal-token slices) is convenient but not representative of actual release cadence (e.g., quarterly “big” releases plus intermittent patches). The auxiliary “composed” time-series that stitches disparate corpora further departs from realistic evolution. Conclusions about trend linearity and predictability could change under longer, lumpier, or recipe-shifting timelines.”
>
> **Response:** We agree that real-world release cadences are often irregular, with larger jumps interspersed with smaller updates. Our goal, however, is not to replicate the exact distribution of vendor release intervals, but to study how cumulative parameter drift affects the longevity of a fixed patch. In this context, the ordering and magnitude of drift are more consequential than the precise timing between updates.
>
> The observations in Appendix G show that degradation rates remain stable across densities, indicating that patch decay does not depend on the granularity or lumpiness of individual updates. While our datasets do not perfectly replicate real-world release cadences, we observed the same gradual degradation trend on two very different datasets, which helps support the generality of our results.
>
> >“Can you expand to production-relevant workloads (code, tool-use, long-context, safety), and replace/augment BLEU for math with verifiable-correctness metrics (programmatic verifiers, unit tests, GSM-style exact correctness)? This would test whether drift rates and the decision tools remain calibrated on harder-to-game metrics.”
>
> **Response:** We thank the reviewer for the helpful suggestions about tasks and assessment metrics. As an initial step toward verifiable evaluation, we trained a patch on GSM8K and report flexible extract answer accuracy across 12 continual pretraining steps. Accuracy declines steadily with a predictable degradation pattern similar to our earlier tasks. This shows that the observed patch degradation pattern persists even under strict, verifiable correctness metrics.
>
> |Time Step|0|1|2|3|4|5|6|7|8|9|10|11|12|
> |---|---|---|---|---|---|---|---|---|---|---|---|---|---|
> Accuracy|0.572|0.255|0.195|0.134|0.129|0.131|0.14|0.114|0.111|0.137|0.128|0.121|0.108|

---

> ### Author Response · Authors · 2025-11-21
> **Response Part 3**
>
> >“Beyond the qualitative explanation, can you measure representational/optimization drift—e.g., CKA similarity across layers, low-rank subspace angles for patched vs. re-tuned adapters, curvature/Fisher changes, or gradient alignment—to link measurable misalignment to observed performance decay? This would strengthen causal plausibility and might reveal layers/subspaces where patches are most robust.”
>
> **Response:** We thank the reviewer for the suggestion about quantifying patch misalignment. To address these concerns, we (i) performed a theoretical analysis to model patch degradation and (ii) performed CKA similarity experiments after receiving the reviewer’s comment.
>
> **(i) Theoretical analysis:**
> Here, we show that the loss in performance due to PortLLM patching can be analyzed in terms of the difference between the PortLLM patch trained at $t=0$, denoted as $\Delta\theta_0$, and the “true” patch trained on the updated model at time $t$, denoted as $\Delta\theta_t$. Full analysis is provided in Appendix J in the revised manuscript.
>
> Let the test loss on a downstream task with parameters $\theta$ be denoted as $\ell(\theta)$. We can then denote performance degradation as
>
> $e(t)=\ell(\theta_t + \Delta\theta_0) - \ell(\theta_t + \Delta\theta_t)$
>
> where $\theta_t$ denote base model parameters at time $t$. Here, the first term denotes loss using the PortLLM paradigm with the $t=0$ patch and the second term denotes periodic repatching loss. Degradation occurs as $\ell(\theta_t + \Delta\theta_0)$ increases as the patch and the base model become more misaligned. Using a Taylor series expansion, we can approximate $e(t)$ as
> $e(t) \approx (\Delta_t)^T \nabla \ell(\theta_t’) + (\Delta_t)^T \nabla^2 \ell(\theta_t’) (\Delta_t) $
> where $\Delta_t=\Delta\theta_0-\Delta\theta_t$ and $\theta_t’=\theta_t+\Delta\theta_t$. Therefore, we can approximate performance degradation in terms of $\Delta_t=\Delta\theta_0 - \Delta\theta_t$, or the difference between the PortLLM patch and the patch that would be trained on the model base model at time $t$. As $\Delta\theta_0 - \Delta\theta_t$ grows in magnitude, the base model and the PortLLM patch become more misaligned and patch performance degrades.
>
> **Our response is continued in the next part below.**

---

> ### Author Response · Authors · 2025-11-21
> **Response Part 4**
>
> >“Beyond the qualitative explanation, can you measure representational/optimization drift—e.g., CKA similarity across layers, low-rank subspace angles for patched vs. re-tuned adapters, curvature/Fisher changes, or gradient alignment—to link measurable misalignment to observed performance decay? This would strengthen causal plausibility and might reveal layers/subspaces where patches are most robust.”
>
> **Response (Continued):**
>
> **(ii) CKA Similarity:**
> To provide complementary empirical evidence of drift, we computed layerwise CKA similarity between the base model at $t_0$ and each continually pretrained checkpoint. We evaluated CKA on inputs drawn from Reddit, BoolQ, MathGenie, and WinoGrande, on both Mistral-7B and Gemma3-4B. Across all datasets and architectures, layers exhibit growing representational changes, consistent with increasing mismatch between the fixed $t_0$ patch and the evolving base model. Representative values for selected layers are shown below; full layerwise curves across all timesteps will be included in the updated manuscript version.
>
> The tables report layerwise CKA similarity between the base model at $t_0$ and later checkpoints across 12 continual pretraining steps. As time progresses, deeper layers show reduced similarity values, reflecting progressively larger representational drift. Downstream patches depend primarily on representations produced in deeper layers [1][2], so drift concentrated in those deeper layers directly affects patch alignment. This change in representation provides a quantitative explanation for why the fixed $t_0$ patch becomes increasingly mismatched with the evolving base model and exhibits predictable performance degradation.
>
> **Reddit Mistral-7B**
> |Time Step|1|2|3|4|5|6|7|8|9|10|11|12|
> |---|---|---|---|---|---|---|---|---|---|---|---|---|
> |Layer 6|1.0|1.0|1.0|1.0|1.0|1.0|1.0|1.0|1.0|1.0|1.0|1.0|
> |Layer 15|0.998|0.996|0.996|0.996|0.996|0.996|0.997|0.996|0.997|0.997|0.997|0.997|
> |Layer 25|0.991|0.978|0.979|0.978|0.981|0.98|0.986|0.984|0.986|0.986|0.985|0.988|
> |Layer 32|0.991|0.969|0.969|0.971|0.98|0.979|0.991|0.988|0.992|0.992|0.991|0.99|
>
> **Reddit Gemma3-4B**
> |Time Step|1|2|3|4|5|6|7|8|9|10|11|12|
> |---|---|---|---|---|---|---|---|---|---|---|---|---|
> |Layer 6|0.999|0.998|0.999|0.996|0.978|0.994|0.991|0.994|0.997|0.991|0.991|0.988|
> |Layer 15|0.959|0.909|0.960|0.937|0.923|0.945|0.925|0.889|0.922|0.933|0.884|0.872|
> |Layer 25|0.967|0.946|0.964|0.959|0.950|0.941|0.954|0.932|0.938|0.944|0.930|0.927|
> |Layer 32|0.952|0.951|0.933|0.938|0.940|0.944|0.934|0.942|0.942|0.943|0.942|0.942|
>
> **BoolQ Mistral-7B**
> |Time Step|1|2|3|4|5|6|7|8|9|10|11|12|
> |---|---|---|---|---|---|---|---|---|---|---|---|---|
> |Layer 6|0.999|0.997|0.997|0.996|0.996|0.995|0.995|0.994|0.994|0.992|0.992|0.991|
> |Layer 15|0.997|0.995|0.994|0.994|0.994|0.993|0.994|0.992|0.993|0.991|0.990|0.990|
> |Layer 25|0.993|0.985|0.985|0.985|0.987|0.984|0.988|0.986|0.987|0.986|0.985|0.986|
> |Layer 32|0.953|0.910|0.915|0.934|0.952|0.944|0.976|0.965|0.976|0.977|0.981|0.981|
>
> **BoolQ Gemma3-4B**
> |Time Step|1|2|3|4|5|6|7|8|9|10|11|12|
> |---|---|---|---|---|---|---|---|---|---|---|---|---|
> |Layer 6|0.997|0.995|0.992|0.971|0.935|0.961|0.95|0.967|0.985|0.972|0.973|0.968|
> |Layer 15|0.976|0.918|0.918|0.909|0.882|0.929|0.872|0.856|0.889|0.902|0.852|0.836|
> |Layer 25|0.957|0.904|0.864|0.894|0.866|0.890|0.880|0.864|0.889|0.853|0.841|0.860|
> |Layer 32|0.943|0.919|0.911|0.904|0.901|0.900|0.899|0.898|0.896|0.893|0.891|0.891|
>
> [1] Chen, G., Yao, Y., Gao, C. J., Chao, L. S., Wan, F., & Wong, D. F. (2025). Not All LoRA Parameters Are Essential: Insights on Inference Necessity. arXiv preprint arXiv:2503.23360.
>
> [2] Houlsby, N., Giurgiu, A., Jastrzebski, S., Morrone, B., De Laroussilhe, Q., Gesmundo, A., Attariyan, M. & Gelly, S.. (2019). Parameter-Efficient Transfer Learning for NLP. Proceedings of the 36th International Conference on Machine Learning

---

> ### Comment · Reviewer_AHDH · 2025-11-26
>
> Thanks for the responses. I will keep the scores unchanged.

---

### Official Review · Reviewer_jCAt · 2025-10-31

**Soundness:** 1
**Presentation:** 3
**Contribution:** 1
**Rating:** 2
**Confidence:** 4

**Summary:**

Frequent updates to base LLMs lead to significant retraining costs for developers who adapt these LLMs to downstream tasks. To address this issue, PortLLM (Khan et al., 2025) proposed a data- and training-free patching method for portability of patches across temporally evolved LLMs. This paper focuses on this particular approach and studies its long-term effectiveness. In particular, the main goal here is to be able to answer this question: How long can PortLLM patching remain effective as the base model evolves?

To answer this, authors perform statistical analysis of patching performance trends across model updates, and develop a time
series modeling framework that characterizes patching performance as a structured temporal process. Based on this modeling framework, authors provide lightweight algorithms to determine when or whether to retrain, without requiring retraining at every base model release.

Experiments were conducted using Mistral-7B model, UpVoteWeb as the continual pretraining corpus and four downstream evaluation datasets.

**Strengths:**

The paper focuses on a practical problem.

The main claims of the paper and the experimental analysis are well presented.

**Weaknesses:**

**Unrealistic experimental setup**
* This work focuses mainly on base LLMs that are evolved via continual pretraining. However, in practice LLM training involves multiple stage/phases (pre/mid/post, SFT/Preference alignment/RL) and practitioners often use post-trained models in real world applications due to their superior instruction following capabilities and human aligned behaviors. Also, for most of the models out there today, the difference between various model versions is rarely as simple as continual pretraining.
* In real world settings, model developers work to make sure that the future model releases are better than earlier model releases. Indeed, in practice, most models get better (on a wide variety of tasks) with successive model releases over time. The continual pretraining setup used this paper (continually pretraining on a small Redditt corpus) is unrealistic as it is not reflecting the realistic setup of base models that improve with time. If we look at Fig. 2 and Fig 27, the base models are getting significantly worse on BoolQ, WinoGrande, ARC-Easy with time and barely improves on MathGenie. Model developers would rarely release such continually degrading models to downstream developers.

**Conclusions that do not extend beyond a particular pretraining corpus**
All the behaviors observed in this paper are strongly tied to the small pretraining corpus used. Continually pretraining on UpVoteWeb makes the base model consistently worse over time which results in the observed (linear/exponential) trends in performance decline. If we use a different corpus (Fig 28 in Appendix), such nice predictable trends may not exist.


**Fairly obvious conclusion**
* In my view, there is nothing surprising or significant in the following statement: "If a model goes through several updates over time, a downstream task patch obtained with first checkpoint will start failing." This paper claims that showing this experimentally as one of their main contributions.

**Questions:**

Authors should use more realistic experimental setup where continual training is not making the base model significantly worse.

---

> ### Author Response · Authors · 2025-11-21
> **Response Part 1**
>
> We thank the reviewer for the careful analysis of our experimental paradigm. We address each concern point by point below.
>
> > “​​The paper focuses on a practical problem.”
> > “This work focuses mainly on base LLMs that are evolved via continual pretraining. However, in practice LLM training involves multiple stage/phases (pre/mid/post, SFT/Preference alignment/RL) and practitioners often use post-trained models in real world applications due to their superior instruction following capabilities and human aligned behaviors.”
> Also, for most of the models out there today, the difference between various model versions is rarely as simple as continual pretraining.
>
> **Response:** Our work is motivated by a practical scenario, but as the first-step exploration in the literature, we simplified the scenario to capture only the dominant factors to ensure the research questions are tractable. Our aim is not to simulate the full complexity of industrial training pipelines, but rather to study how a fixed patch trained at $t_0$ behaves as the base model parameters drift over time. We present the first analysis of long-term patch utility under continual base model updates. For this initial analysis, some simplifications are needed to make the analysis tractable.
>
> We believe that additional methods for updating the base model (e.g., reinforcement learning) do not affect the rate of patch degradation. The RL updates may be viewed as an additional patch applied to the base model's parameters. This patch, which provides safety guardrails for general tasks, could be largely orthogonal to the PortLLM patch for downstream, domain-specific tasks.
>
> > “In real world settings, model developers work to make sure that the future model releases are better than earlier model releases. Indeed, in practice, most models get better (on a wide variety of tasks) with successive model releases over time. The continual pretraining setup used this paper (continually pretraining on a small Redditt corpus) is unrealistic as it is not reflecting the realistic setup of base models that improve with time. If we look at Fig. 2 and Fig 27, the base models are getting significantly worse on BoolQ, WinoGrande, ARC-Easy with time and barely improves on MathGenie. Model developers would rarely release such continually degrading models to downstream developers.”
>
>
> **Response:** Prior work has documented that continual pretraining often introduces a stability gap, where downstream performance initially drops before recovering after substantial additional training. This behavior has also been observed in LLMs [1]. While mitigation strategies exist, we intentionally did not apply them to maintain a controlled, tractable drift process for isolating patch-based misalignment. As we will justify in the response to the next comment, dropping downstream performance of the base model is not a major factor to the phenomenon that this paper studies.
>
> [1] Yiduo Guo, Jie Fu, Huishuai Zhang, and Dongyan Zhao. 2025. Efficient Domain Continual pretraining by Mitigating the Stability Gap. In Proceedings of the 63rd Annual Meeting of the Association for Computational Linguistics (Volume 1: Long Papers), pages 32850–32870, Vienna, Austria. Association for Computational Linguistics.

---

> ### Author Response · Authors · 2025-11-21
> **Response Part 2**
>
> > “All the behaviors observed in this paper are strongly tied to the small pretraining corpus used. Continually pretraining on UpVoteWeb makes the base model consistently worse over time which results in the observed (linear/exponential) trends in performance decline. If we use a different corpus (Fig 28 in Appendix), such nice predictable trends may not exist.”
>
> **Response:** We respectfully disagree that all of our results are tied to the UpVoteWeb corpus. In addition to UpVoteWeb, our initially submitted manuscript also evaluated a composed multi-corpus time series (Figures 9, 10, and 28), which introduces substantially larger distributional shifts and heterogeneous update magnitudes. In Figure 28, we focus on the blue $t_0$ patching trace. We observe gradual performance degradation perturbed by some noise, as predicted by Equation (2). The p-values in Table 4 indicate that our results are statistically significant in 16 out of 22 cases. A detailed discussion of the datasets used in the combined time-series dataset is provided in Appendix I.
>
> To address the reviewer’s concern that the dropping trends of test performance might be driven solely by base-model degradation, we (i) conducted an additional experiment on Gemma3-4B with a base model having a non-monotonic-decreasing trend, and (ii) theoretically analyzed the increase of test error and ruled out the direct impact due to the base-model degradation.
>
> *(i) Additional experiment on Gemma3-4B:*
> We show via additional experiments with Gemma3-4B that patch degradation does not simply mirror degradation of the base model. We continually pretrained Gemma3-4B on UpVoteWeb, and observed that the base model’s accuracy on BoolQ increases at several timesteps (see 1st row below), while the $t_0$-patch exhibits a much more consistent decline (see 2nd row).
>
> |Time Step|0|1|2|3|4|5|6|7|8|9|10|11|12|
> |---|---|---|---|---|---|---|---|---|---|---|---|---|---|
> |BoolQ **No Patch** Performance (Acc)|0.790|0.677|0.647|0.620|0.576|0.513|0.529|0.533|0.591|0.614|0.603|0.613|0.578|
> |BoolQ **Patched** Performance (Acc)|0.894|0.865|0.818|0.759|0.738|0.695|0.698|0.667|0.670|0.653|0.638|0.654|0.612|
>
> This indicates that the phenomenon of patching performance drop still occurs even when the base model improves.
>
> *(ii) Theoretical justification ruling out the direct impact of base-model degradation:*
> We claim that degradation will occur independent of performance on the base model. For a complete theoretical treatment, please refer to Appendix J in the supplemental document. Below, we highlight the key points of the claim.
> Let the test error of a neural network parameterized by $\theta$ over a downstream task be denoted as $\ell (\theta)$. Denote the base model at time $t$ as $\theta_t$ and the performance of the base model on the downstream task $\ell(\theta_t)$. The patch degradation in terms of test error increase is given by
>
> $e(t)=\ell(\theta_t+\Delta\theta_0) - \ell(\theta_t+\Delta\theta_t)$,
>
> where $\Delta\theta_0$ is the suboptimal PortLLM patch and $\Delta\theta_t$ is the “true” patch obtained by fine-tuning over the downstream task on base model $\theta_t$. In the equation above, the first term denotes the PortLLM patch’s testing error using the PortLLM patch obtained at $t=0$, whereas the second term represents the best possible testing error (via periodic repatching at current $t$).
>
> Note that the dropping base-model performance $\ell(\theta_t)$ as $t$ increases, which the reviewer hypothesized that “results in the observed ... trends in performance decline”, depends on a different point (i.e., $\theta_t$ vs. $\theta_t+\Delta\theta_t$ and $\theta_t+\Delta\theta_0$) in the loss landscape than the PortLLM patching degradation $e(t) = \ell(\theta_t+\Delta\theta_t) - \ell(\theta_t+\Delta\theta_0)$. In other words, test error $\ell(\theta_t)$ being a monotonically increasing function of $t$ does not directly imply patch degradation $e(t)$ being a monotonic increasing function. This indicates that degradation in base model performance will not have a substantial impact on patched performance. (Rather, patched performance degrades because of growing misalignment between the base model $\theta_t$ and the patch $\Delta\theta_0$ as $t$ increases, as we showed in the newly added Appendix J of the supplemental document.)

---

> ### Author Response · Authors · 2025-11-21
> **Response Part 3**
>
> > “In my view, there is nothing surprising or significant in the following statement: "If a model goes through several updates over time, a downstream task patch obtained with first checkpoint will start failing." This paper claims that showing this experimentally as one of their main contributions.”
>
> **Response:** We agree that a patch’s degradation over time can be hypothesized, but the extent, shape, and consistency of this degradation cannot be assumed without empirical evidence. PortLLM (Khan et al., 2025) reported that patches remain effective temporally across multiple model updates. Our results show that this is not always the case and that degradation appears earlier, more consistently, and more predictably than previously understood. Our evaluations also reveal that degradation follows a smooth, measurable temporal trajectory. Without large-scale longitudinal experiments, one could equally hypothesize sudden failure modes, oscillatory behaviors, or recovery phases. The empirical finding that degradation is gradual and statistically predictable is therefore nontrivial. In addition to the empirical analysis, our contribution includes a modeling framework that characterizes patch performance as a structured temporal process, along with two forecasting algorithms that provide actionable tools for real-world deployment scenarios.

---

### Meta-Review · Area_Chair_YWBT · 2026-01-01

**Summary:**

This paper studies the temporal robustness of PortLLM-style training-free patches under continual base-model evolution, proposing statistical models and forecasting tools to decide when repatching is needed. Reviewers broadly agreed that the problem is practically motivated and clearly framed, and several appreciated the longitudinal evaluation pipeline and decision-oriented tools.

However, the dominant concerns across reviews center on external validity and realism. Two reviewers (jCAt, AHDH) were unconvinced that the experimental setup, i.e., continual pretraining via LoRA on UpVoteWeb slices, adequately reflects real-world LLM evolution involving heterogeneous training phases, alignment steps, and improving model quality over time. There was skepticism that the observed smooth degradation trends generalize beyond this controlled setting or a single model lineage. Another recurring concern was that the core empirical finding (patches degrade over time) may be intuitive, with insufficient evidence that the proposed modeling framework yields fundamentally new insights beyond confirming this expectation.

**Reviewer Concerns:**

Concerns Largely Addressed by the Rebuttal
- Generality beyond a single model family:
  Additional Gemma3-4B experiments (BoolQ, WinoGrande) convincingly showed similar degradation trends across architectures (addressing AHDH, partially jCAt).
- Base-model degradation vs. patch misalignment:
  Theoretical analysis and non-monotonic base-model results (Gemma3-4B) clarified that patch decay is not a trivial mirror of base-model performance drops (jCAt).
- Mechanistic explanation:
  Added theoretical derivations and CKA similarity analyses provided a concrete account of patch–base misalignment over time (AHDH).
- Evaluation rigor:
  Inclusion of GSM8K with verifiable correctness addressed concerns about metric gaming and strengthened empirical credibility (AHDH).
- Actionability of forecasting tools:
  Reviewer S9hM acknowledged that most concerns were addressed and maintained a positive score.

Concerns Still Outstanding
- Realism of upstream evolution:
  Continual pretraining via LoRA on curated slices remains a weak proxy for heterogeneous, vendor-style model updates (full-parameter training, RLHF, recipe changes). This concern remained decisive for jCAt and AHDH.
- External validity and scope:
  Evidence is still limited to PortLLM-style patches and a narrow set of downstream tasks; no toy or comparative study convincingly demonstrates applicability to other patching/editing paradigms.
- Strength of contribution relative to intuition:
  Some reviewers remained unconvinced that demonstrating gradual, predictable degradation constitutes a sufficiently non-obvious or high-impact contribution for ICLR, even with added modeling.
- Temporal scale and release cadence:
  Short, evenly spaced timelines may not capture longer-term or irregular real-world release dynamics.

**Reviewer Scores:**

- jCAt
  - Original Score: 2 (reject)
  - Estimated Post-Discussion Score: 2
  - Rationale: Core realism and external validity concerns remained.

- AHDH
  - Original Score: 2 (reject)
  - Estimated Post-Discussion Score: 2
  - Rationale: The reviewer explicitly stated that their scores remained unchanged despite the additional experiments and analysis in the rebuttal.

- S9hM
  - Original Score: 6 (marginal accept)
  - Estimated Post-Discussion Score: 6
  - Rationale: The rebuttal addressed most concerns; the reviewer maintained a positive score with only a minor remaining question.

- fxQe
  - Original Score: 6 (marginal accept)
  - Estimated Post-Discussion Score: 6
  - Rationale: The authors largely acknowledged and addressed the reviewer’s concerns, with no indication that the reviewer would downgrade their score.

---

### Decision · Program_Chairs · 2026-01-26

Reject